 **eLIFE**

# Selection of distinct populations of dentate granule cells in response to inputs as a mechanism for pattern separation in mice

**Wei Deng[1], Mark Mayford[2], Fred H Gage[1]\***

[1]Laboratory of Genetics, The Salk Institute for Biological Studies, La Jolla, United States; [2]Department of Cell Biology and Dorris Neuroscience Center, The Scripps Research Institute, La Jolla, United States

**Abstract** The hippocampus is critical for episodic memory and computational studies have predicted specific functions for each hippocampal subregion. Particularly, the dentate gyrus (DG) is hypothesized to perform pattern separation by forming distinct representations of similar inputs. How pattern separation is achieved by the DG remains largely unclear. By examining neuronal activities at a population level, we revealed that, unlike CA1 neuron populations, dentate granule cell (DGC) ensembles activated by learning were not preferentially reactivated by memory recall. Moreover, when mice encountered an environment to which they had not been previously exposed, a novel DGC population—rather than the previously activated DGC ensembles that responded to past events—was selected to represent the new environmental inputs. This selection of a novel responsive DGC population could be triggered by small changes in environmental inputs. Therefore, selecting distinct DGC populations to represent similar but not identical inputs is a mechanism for pattern separation.

**\*For correspondence:** gage@salk.edu

**Competing interests:** The authors declare that no competing interests exist.

## Introduction

During learning and memory, the hippocampus, a key structure for episodic memory, receives information from the cortex through multiple parallel pathways to each of its main subregions, including the dentate gyrus (DG), CA3 and CA1 (*Squire, 1992*; *Rolls and Kesner, 2006*; *Rolls, 2010*). The DG receives excitatory inputs from entorhinal cortex (EC) layer II neurons via the perforant pathway and relays the information to CA3 through mossy fibers. The CA3 in turn projects to CA1, which sends back-projections to deep layers of the EC, forming the classic tri-synaptic pathway (EC→DG→CA3→CA1). CA3 also receives direct inputs from EC through the perforant pathway and there are extensive interconnections among CA3 neurons via recurrent collateral fibers. In addition to inputs from CA3, CA1 receives inputs directly from EC layer III neurons through the temporoammonic pathway, forming a monosynaptic pathway (EC→CA1). In this complex neural network, each pathway and each subregion is likely to carry out specific functions during learning and memory.

Based on these network connections and the anatomical characteristics of each subregion, theories about specific functions of the individual hippocampal subregions in learning and memory have been proposed by computational modeling (*Rolls and Kesner, 2006*; *Rolls, 2010*). In particular, the DG is postulated to function as a pattern separator by de-correlating inputs from EC (*Marr, 1971*) because of its sparse activity and its considerably larger population of neurons compared to the EC and CA3. The pattern separation function of the DG is supported by accumulating evidence from behavioral studies, reporting that animals with lesions or blocked plasticity in the DG were impaired in discriminating similar spatial and contextual information (*Gilbert et al., 2001*; *McHugh et al., 2007*; *Goodrich-Hunsaker et al., 2008*; *Nakashiba et al., 2012*). Nevertheless, how the DG achieves the pattern separation

**eLife digest** Being able to keep memories of similar events separate in your mind is an essential part of remembering. If you use the same carpark every day, recalling where you left your car this morning is challenging, not because you have to remember an event from long ago, but because you have to distinguish between many similar memories.

Keeping memories distinct is one of the functions of a subregion of the hippocampus called the dentate gyrus. The process of taking complex memories and converting them into representations that are less easily confused is known as pattern separation. Exactly how the dentate gyrus achieves this, however, is unclear.

Computational models predict that a different population of dentate gyrus cells will be active when an animal is in different environments. However, previous experiments have instead shown that the same population of cells is active in multiple environments, and that cells distinguish between environments by firing at different rates.

Now, Deng et al. have added to our understanding of pattern separation. The researchers used a type of genetically modified mouse in which it is possible to identify or 'tag' the activity of a population of hippocampal neurons at multiple time points. They placed each mouse in a box and noted which hippocampal neurons were active as the animal learned about its new environment. After several such learning episodes, the animal received a mild electric shock inside the box. When it was returned to the box the next day, the mouse remembered receiving the shock, enabling the researchers to note which neurons were active during the retrieval process.

Deng et al. found that in a subregion of the hippocampus called CA1, the particular neurons that were active during the initial learning episode were also likely to be active when the animals remembered receiving the shock. However, this was not the case for the dentate gyrus: in this subregion, distinct groups of cells were active during learning and during retrieval. Moreover, exposing the mice to two subtly different environments activated two distinct groups of cells in the dentate gyrus.

The work of Deng et al. reveals that memory retrieval does not always involve reactivation of the same neurons that were active during encoding. More importantly, the results indicate that the dentate gyrus performs pattern separation by using distinct populations of cells to represent similar but non-identical memories. Overall the findings add to our understanding of the mechanisms that underpin memory formation.

function remains elusive. In vivo physiological recordings of dentate granule cells (DGCs) have shown that changes in environmental inputs only evoke the rate remapping of DGCs but not the global remapping predicted by computational models (*Leutgeb et al., 2007*). Through the powerful mossy fiber synapses, outputs of the DG are passed to the downstream recurrent network in CA3, which is hypothesized to be the site for memory storage (*Treves and Rolls, 1994*). Computational studies have suggested that it is advantageous to have two extrinsic afferent systems for the autoassociative network in CA3—one with strong synapses for memory formation and the other with associatively modifiable synapses for memory retrieval (*Treves and Rolls, 1992*). Therefore, it has been speculated that the mossy fiber inputs from the DG may be particularly suitable for memory formation, whereas the direct inputs from EC may be responsible for information recall. On the other hand, CA1 is considered to be a feed forward neural network and is the main output region for the hippocampus (*Rolls and Kesner, 2006*; *Rolls, 2010*). Experimental evidence from genetic and physiological studies has demonstrated the importance of CA1 for both memory formation and retrieval (*Riedel et al., 1999*; *Dupret et al., 2010*; *Goshen et al., 2011*). Because the large size of the DGC population is a key factor for the computational hypothesis of pattern separation, we utilized TetTag transgenic mice to examine the population neuronal activity of the dorsal DG to test whether DGCs undergo global remapping at the population level. To examine the specificity of the responsiveness of the DGCs, population activity in CA1 was also analyzed. Our results revealed a novel mechanism for pattern separation in the DG: the selection of distinct DGC populations to represent different contextual information. In addition, we observed that memory recall preferentially reactivated the neuronal population involved in learning in CA1 but not in the DG, suggesting that, in a complex neural network, memory recall may not reinstate the activities in every pathway involved in memory formation.

## Results

### Using TetTag transgenic mice to examine neuronal activity at a population level

We studied the population activity of neurons in the hippocampus by examining the transient expression of immediate early genes (IEGs, such as *Fos*, *Arc* and *Egr1*), which is commonly used as an indicator of recent neuronal activity (*Guzowski et al., 2005*). To compare the activities in the same neuronal population in response to two events at sequential time points, we used TetTag bi-transgenic mice in which neuronal activities at a given time window can be persistently labeled (*Figure 1A*, *Figure 1—figure supplement 1*; *Reijmers et al., 2007*). In these mice, neuronal activity can activate the *Fos* promoter and induce the expression of tetracycline-controlled transactivator (tTA) from the *Fos-tTA* transgene. In the absence of doxycycline (dox), a drug that binds to tTA and prevents tTA from binding to the tetracycline responsive promoter (*tetO*), the resulting tTA can activate the expression of the tau-LacZ marker from the transgene: *tetO-tau-lacZ:tTA\**. At the same time, a tetracycline-insensitive form of transactivator (tTA\*: tTA containing H100Y point mutation) is also expressed, allowing the persistent tau-LacZ expression irrespective of dox treatment. Thus, if the mice are removed from dox treatment for an initial experience and euthanized shortly after a second experience, the activity of the same neuronal ensemble in response to these two sequential experiences can be assessed by examining the expressions of tau-LacZ and IEGs, which correspond to neuronal activities of the first and second experiences, respectively.

First, we tested whether expression of tau-LacZ markers in the hippocampus of TetTag mice could be regulated by dox. We exposed mice to an enriched environment under either a dox-on or dox-off condition (*Figure 1B,E*, see 'Materials and methods') and found that removing dox treatment effectively induced tau-LacZ expression in neurons of the DG and CA1 (*Figure 1B–G*, *Figure 1—figure supplement 2A–B*), with most of the LacZ-positive neurons displaying typical morphologies of granule cells and pyramidal neurons in the DG and CA1, respectively (*Figure 1F,G*). Furthermore, many LacZ-positive cells also co-expressed FOS, with over 70% and 85% of LacZ-positive cells expressing FOS in the DG and CA1, respectively, suggesting that the expression of LacZ did not affect the expression of IEGs in the same neuron (*Figure 1F,G*, *Figure 1—figure supplement 3A*). It was also notable that the efficiency of tagging was low, compared to the endogenous FOS labeling, particularly in the CA1 region. A low efficiency of tagging was also observed in basolateral amygadala (*Reijmers et al., 2007*). This low and variable induction efficiency across brain regions was possibly caused by low penetrance and variable expressivity of the transgenes, a common problem for transgenic mice. To test if the tagged population represents activities in the general population, we measured the intensity of FOS staining in the LacZ-positive and LacZ-negative neurons. In both DG and CA1, the FOS intensity was similar between LacZ-positive and LacZ-negative populations (*Figure 1—figure supplement 3B–E*). Therefore, it was likely that LacZ tagged neurons were representatives of the activated population, although we could not formally rule out the possibility that only a specific population of activated neurons (e.g. the population with the highest activities) could be tagged. The induction efficiency was even lower in CA3 with few neurons tagged (*Figure 1—figure supplement 2*), preventing further analysis in this region.

We next tested the activity-dependent expression of tau-LacZ markers in TetTag mice. After removing them from dox treatment, we exposed some mice to a fear conditioning chamber (ctxA, *Figure 2—figure supplement 4*) and kept others in their home cage (HC) ('Materials and methods'). While LacZ-positive neurons could be readily detected in both CA1 and the DG in the HC mice (*Figure 2A,B*), substantially more LacZ tagged neurons were observed in the ctxA mice in both CA1 and the DG (*Figure 2C–F*; t-test, in CA1, HC, 1.5 ± 0.5%, n = 4; ctxA, 5.1 ± 0.5%, n = 3; p<0.007; in the DG, HC, 1.9 ± 0.7%; ctxA, 6.9 ± 1.0%; p<0.016). Therefore, the dox-regulated and activity-dependent expression of LacZ in both the DG and CA1 suggested the feasibility of studying neuronal activities at a population level in these hippocampal subregions using TetTag mice.

### Preferential reactivation of CA1 neuronal population involved in learning by memory recall

To study the activity of neuronal populations in the DG and CA1 during event learning and subsequent memory recall, we used a contextual fear conditioning paradigm combining contextual pre-exposure and immediate foot shock (*Fanselow, 1990*, *2000*, *2010*). This is a task in which the hippocampus has

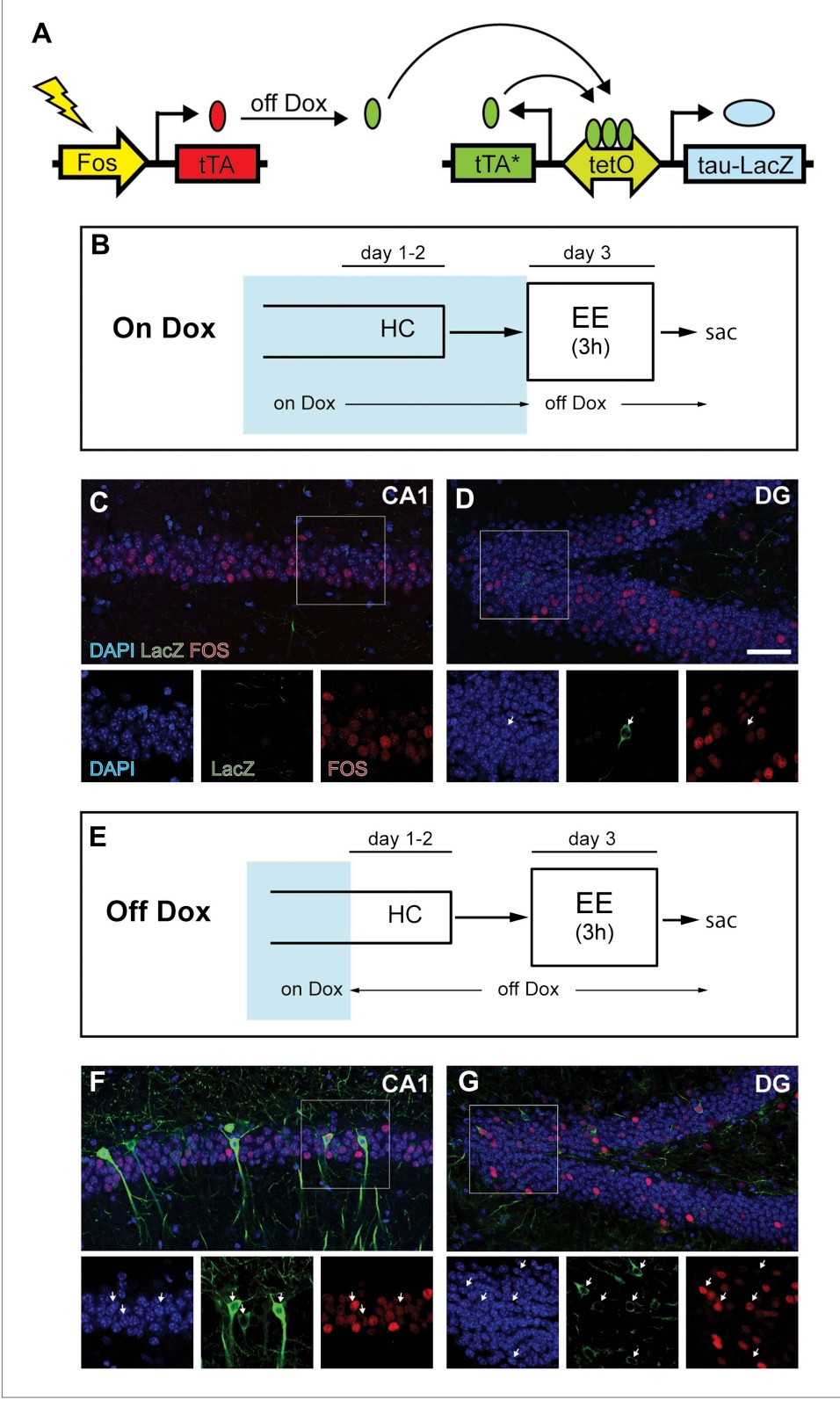

**Figure 1**. Induction of tag (tau-LacZ) expression by removing dox treatment. (**A**) A brief cartoon illustrating the TetTag transgenic system. (**B**) and (**E**) Experimental designs. Dox treatment is illustrated by blue shading. (**C**) and (**D**) There are few neurons in either CA1 (**C**) or the DG (**D**) (outlined by DAPI [blue]) expressing LacZ marker (green) if mice are kept on a dox diet until enriched environment (EE) exposure. Samples are also stained with FOS (red). *Figure 1. Continued on next page*

*Figure 1. Continued*

Each channel in the inset (outlined by the square) is presented below the corresponding overall image, with arrows indicating the LacZ-positive neurons. (**F**) and (**G**) In mice that were removed from dox treatment 2 days before EE exposure, many LacZ-positive neurons can be observed in both CA1 (**F**) and the DG (**G**). In both subregions, many tagged neurons are also co-stained with FOS. The scale bar in (**D**) represents 50 μm for (**C**, **D**, **F**, and **G**).

The following figure supplements are available for figure 1:

**Figure supplement 1**. The TetTag system.

**Figure supplement 2**. Induction of tau-LacZ expression in the hippocampus by removing mice from dox treatment.

**Figure supplement 3**. Quantification of activities and FOS intensity in mice exposed to an enriched environment during the dox-off window.

been demonstrated to be critically involved in forming a conjunctive representation of the conditioning context during pre-exposure (*Barrientos et al., 2002*; *Rudy et al., 2002*; *Stote and Fanselow, 2004*). We chose this task because the formation of the contextual memory, which is dependent on the hippocampus, can be temporally separated from the subsequent context-shock association, which presumably relies mostly on the function of amygdala (*Rudy and O'Reilly, 2001*; *Rudy et al., 2002*; *Reijmers et al., 2007*; *Han et al., 2009*). With dox treatment removed, we pre-exposed one group of mice (preA, n = 12) to the fear conditioning chamber (context A) to tag the activated neurons (LacZ+) in contextual learning (*Figure 3A*, *Figure 2—figure supplement 1*). After the last pre-exposure (on day 5), mice were put back on dox treatment to prevent further tagging. 2 days later, mice were subjected to immediate shock in the conditioning chamber and their conditioned fear memory was tested 1 day after immediate shock. Mice were perfused shortly after the memory test for neuronal activity analysis. For comparison, another group of mice (preC, n = 11) was subjected to the identical protocol except that they were pre-exposed to an environment (context C, *Figure 2—figure supplement 1*) that was completely different from the conditioning chamber. Because the mice associated contextual information during pre-exposure with the subsequent aversive stimulus (i.e. foot shock) in this protocol, it was not surprising that preA mice but not preC mice displayed a high level of freezing behavior when the mice were tested for their conditioned response in context A (*Figure 3B*; t-test, $t_{21} = 3.424$, p<0.0026).

To investigate the activities in neuron populations of the DG and CA1, we concurrently examined the expression of tau-LacZ and the expression of IEGs to evaluate the neuronal activities during contextual pre-exposure and during memory recall test, respectively (*Figure 3—figure supplement 1*, see 'Materials and methods'). We focused our analysis on the dorsal hippocampus, because this region has been shown to be tightly associated with learning and memory. For technical convenience, FOS and EGR1 were used as markers to assess the recall-activated neurons in CA1 and the DG, respectively, and we designated the percentage of IEG positive neurons in the total numbers of neurons quantified as the activation rate ('Materials and methods'). To measure the proportion of the neurons that were activated by the recall test in the neuronal population that was previously activated during pre-exposure, we quantified the percentage of LacZ+IEG double positive neurons in the LacZ tagged population (designated as the reactivation rate).

We were not able to detect a significant difference in either CA1 or the DG in the percentage of LacZ positive neurons in the total numbers of neurons quantified between preA and preC mice (*Figure 3C,D*; t-test, CA1: $t_{21} = 0.5005$, p>0.62; DG: $t_{21} = 0.8504$, p>0.40), suggesting that contexts A and C had equivalent simulating effects. To investigate how the neurons involved in memory formation responded to subsequent memory recall, we compared the reactivation rates to the corresponding activation rates. In CA1, whether or not the neurons that were activated during pre-exposure were preferentially activated again by the recall test in context A depended on the identity of the pre-exposure context (*Figure 3E*; ANOVA: group x activity rates interaction, $F_{1,1} = 11.60$, p<0.0027; main effect of activity rates, $F_{1,21} = 44.04$, p<0.0001; main effect of group, $F_{1,21} = 8.238$, p<0.0092). Because both groups of mice were tested in context A, there was no significant difference in the activation rate

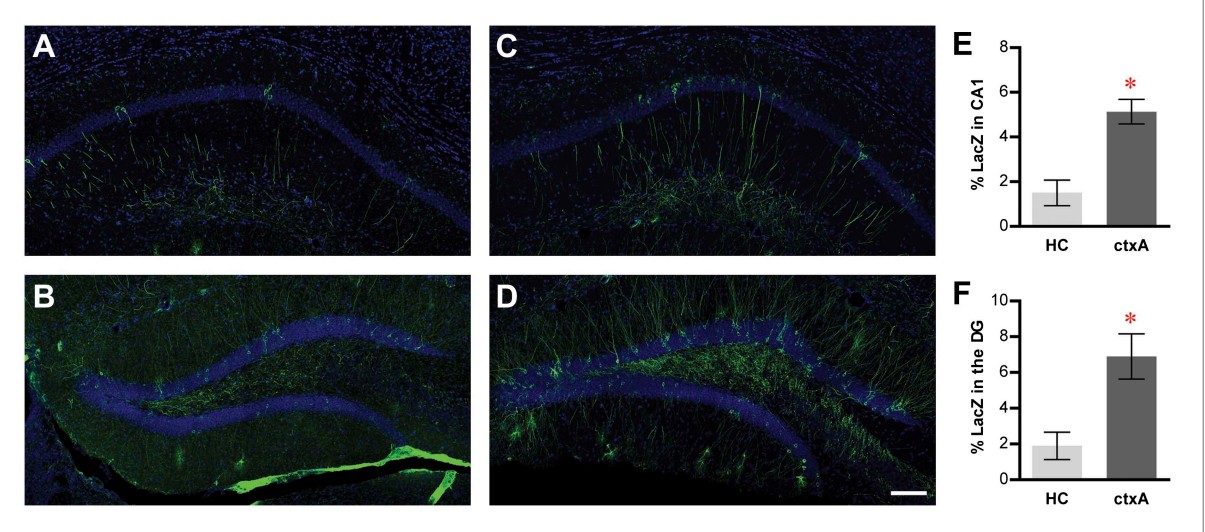

**Figure 2**. Activity-dependent induction of tag (tau-LacZ) expression. (**A**) and (**B**) The expressions of LacZ marker (green) in CA1 (**A**) and DG (**B**) of the mice in the home cage (HC) group. The overall anatomies are highlighted by the DAPI staining (blue). (**C**) and (**D**) The expressions of LacZ marker in CA1 (**C**) and the DG (**D**) of the mice in the context A (ctxA) group. (**E**) and (**F**) Quantification demonstrates that the numbers of LacZ-positive neurons are significantly higher in the ctxA group compared to the HC group in both CA1 (**E**) and the DG (**F**). The scale bar in (**D**) represents 100 µm for panels (**A–D**). Asterisk indicates statistically significant difference between groups. Data are shown as mean ± SEM.

The following figure supplements are available for figure 2:

**Figure supplement 1**. Contexts used for contextual fear conditioning.

between preA and preC mice, as expected (***Figure 3E***, Bonferroni post hoc test, p>0.05). In contrast, the reactivation rate of preA mice was significantly higher than that of preC mice (***Figure 3E***, Bonferroni post hoc test, p<0.001), indicating that a previous learning experience affected neuronal responses at the time of memory recall. In preA mice, which underwent pre-exposure and retested in context A, the reactivation rate was significantly higher than the activation rate (***Figure 3E***, Bonferroni post hoc test, p<0.0001), suggesting that CA1 neurons that were activated during learning were preferentially reactivated by subsequent memory recall. By contrast, in preC mice, neurons responding to context C during pre-exposure were not preferentially activated by the subsequent test in context A (***Figure 3E***, Bonferroni post hoc test, activation rate vs reactivation rate in preC mice, p>0.05), suggesting that recall-induced preferential reactivation of the CA1 neuron population involved in memory formation depended on retrieval of the same memory trace. We further quantified the degree of this reactivation preference by a reactivation index, which normalized the reactivation rate by the corresponding activation rate ('Materials and methods'). In CA1, the reactivation index in preA but not preC mice was significantly above chance (***Figure 3G***; one sample t-test, chance = 0: preA, $t_{11}$ = 12.12, p<0.0001; preC, $t_{10}$ = 1.337, p>0.20), with the index of preA mice being significantly higher than that of preC mice (***Figure 3G***; t-test, $t_{21}$ = 3.115, p<0.0048). These data allow us to propose that the CA1 neuronal ensemble that is responsible for contextual learning is likely reinstated for the recall of the same memory trace. The consistency of this finding with previous reports that showed that CA1 is involved in both memory formation and retrieval (***Riedel et al., 1999***; ***Goshen et al., 2011***) further validates our methodology of using TetTag mice for population neuronal activity study in the hippocampus.

### Selection of distinct populations of DGCs to represent different events in the DG

In contrast to CA1, memory recall did not induce the preferential reactivation of the population of DGCs that was activated during learning, as indicated by the similar activation rate and reactivation rate in preA mice (***Figure 3F***; ANOVA: group x activity rates interaction, $F_{1,1}$ = 18.51, p<0.0003; no

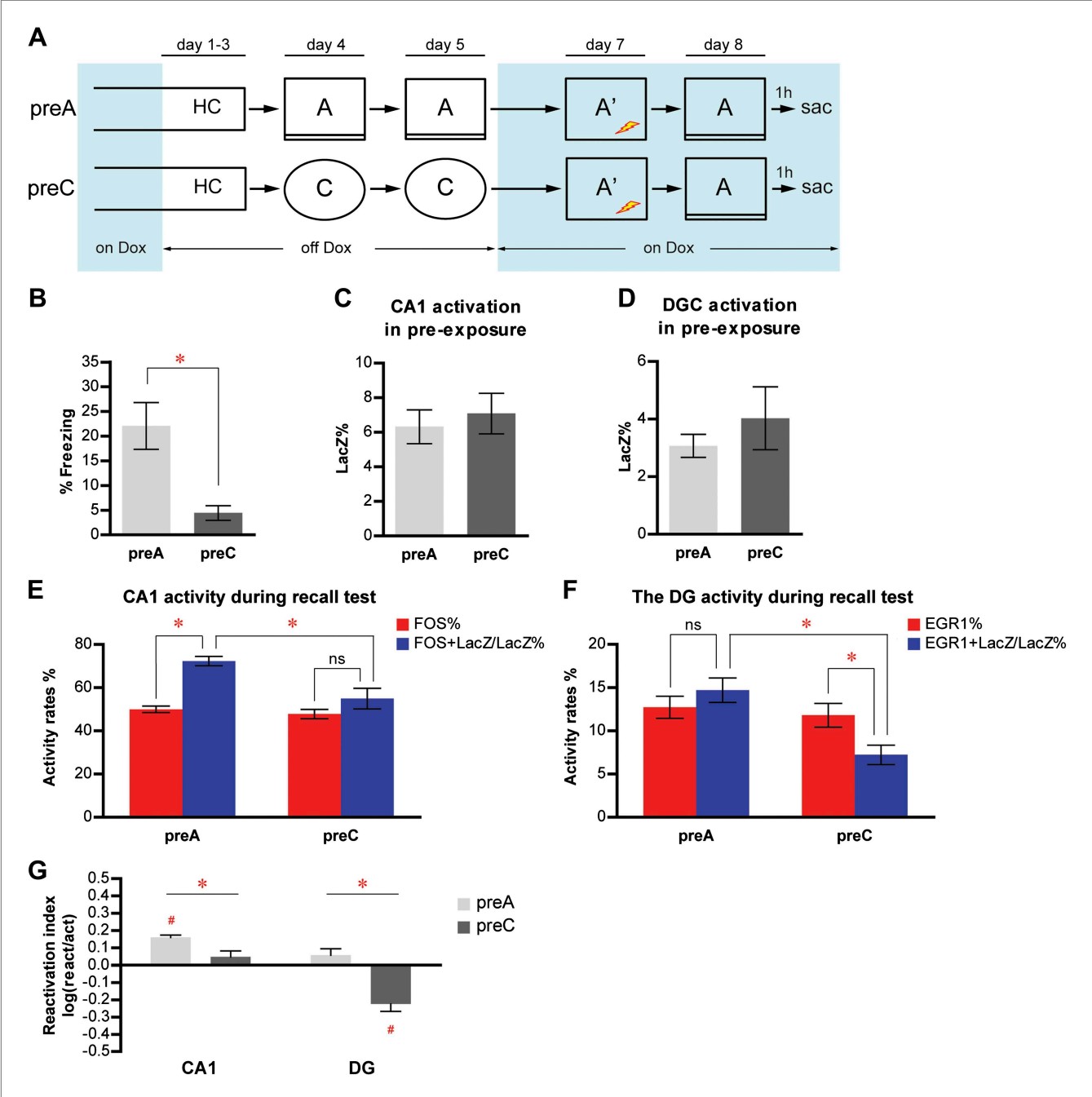

**Figure 3**. Memory recall preferentially reactivates the neuron population in response to learning in CA1 but not in the DG. (**A**) The pre-exposure-immediate shock paradigm for contextual fear conditioning. Dox treatment is illustrated by blue shading. Contextual learning mainly takes place during pre-exposure in the absence of dox treatment. LacZ and IEGs (FOS or EGR1) are regarded as indicators of learning-induced activity and retrieval-induced activity, respectively. Dox treatment is illustrated by blue shading. (**B**) preA mice display significantly more freezing behavior than preC mice. (**C**) and (**D**) During pre-exposure, the proportions of LacZ-positive neurons in either CA1 (**C**) or the DG (**D**) are not significantly different between preA and preC mice. (**E**) During the retrieval test, preferential reactivation of the LacZ-positive population in CA1 is revealed by quantifying the percentage of FOS-positive neurons in the total population (activation rate) and the percentage of LacZ-FOS double-positive cells in the LacZ-positive population (reactivation rate). (**F**) There is no preferential reactivation of LacZ-positive DGCs in preA mice, whereas LacZ-positive DGCs are significantly less likely to be reactivated in preC mice compared to preA mice. The reactivation rate is not significantly different from the activation rate in preA mice but is significantly lower than the activation rate in preC mice. (**G**) Reactivation indexes suggesting the differential reactivations of learning-induced neuronal ensembles by recall in CA1 and the DG (ANOVA: region x group interaction, $F_{1,1}$ = 5.016, p<0.037; main region effect, $F_{1,21}$ = 24.49, p<0.0001; main group

*Figure 3. Continued on next page*

*Figure 3. Continued*

effect, $F_{1,21}$ = 50.10, p<0.0001). Asterisk indicates statistically significant difference between groups. Hash indicates statistically significant difference from chance. Data are shown as mean ± SEM (ns: no significant difference; HC: home cage; sac: sacrifice).

The following figure supplements are available for figure 3:

**Figure supplement 1**. Representative confocal images illustrating the expression of IEGs and LacZ in CA1 (tau-LacZ in green, FOS in red, RBFOX3 in blue) and the DG (tau-LacZ in green, EGR1 in red, DAPI in blue).

**Figure supplement 2**. Quantification of the entire z-series of confocal images in the DG for the contextual fear conditioning experiment 1.

main effect of activity rates, $F_{1,21}$ = 2.910, p=0.1028; main effect of group, $F_{1,21}$ = 6.213, p<0.022; Bonferroni post hoc test, activation rate vs reactivation rate in preA, p>0.05). To our surprise, in preC mice, the reactivation rate of DGCs was significantly lower than the activation rate (*Figure 3F*; Bonferroni post hoc test, p<0.001), indicating that the DGC population responding to context C was significantly less likely to be activated by context A compared to the general DGC population. Compared to activation rates that were not significantly different between preA and preC mice (*Figure 3F*; Bonferroni post hoc test, p>0.05), the reactivation rate of preC mice was significantly lower than that of preA mice (*Figure 3F*; Bonferroni post hoc test, p<0.001). Thus, rather than the DGC population responding to context C, preC mice activated a different population of DGCs in response to context A. These results were further confirmed by the analysis of reactivation indexes. The reactivation index was significantly higher in preA mice compared to that of preC mice (*Figure 3G*; t-test, $t_{21}$ = 5.032, p<0.0001), with the index in preC but not preA mice significantly below chance (*Figure 3G*; one sample t-test, chance = 0: preA, $t_{11}$ = 1.550, p>0.14; preC, $t_{10}$ = 5.314, p<0.0003). To substantiate these results, we re-analyzed the data in the DG by quantifying the activities in the entire z-series of confocal images and obtained similar results (see 'Materials and methods', *Figure 3—figure supplement 2*). Given that the quantification of the entire z-series increased the sampling size, all subsequent analyses were carried out using this approach. Furthermore, we re-measured the activation and reactivation rates in a subset of preA and preC mice, using the expression of FOS as the indicator for DGC activities in the recall test. Similar results were obtained using either FOS or EGR1 as activity indicators in the same cohort of mice (*Figure 4*). In summary, these analyses of the population activities of DGCs demonstrated that neurons in the DG and CA1 responded differently during memory processing. Unlike CA1 pyramidal neurons, DGCs activated by learning an event were not preferentially reactivated by retrieving the same memory. Instead, distinct ensembles of DGCs were selected in response to different events.

To determine how these results could be affected by HC activity, an inevitable part of both pre-exposing and re-exposing experiences, and whether the emotional value of the learned context was critical for population reactivation, we performed a new experiment with two modifications of the previous procedures. First, one group of mice (HC mice, n = 4) were kept in their HC without exposure to any context during the dox-off window while the other group was exposed to context A (ctxA, n = 7); second, the immediate-shock procedure was omitted so that the pre-exposed context remained emotionally neutral for animals at the re-exposure (*Figure 5A*). After all mice were put back on dox treatment, the HC mice were subsequently re-exposed to context A, whereas ctxA mice were further divided into two groups and re-exposed to either context A (ctxA/A, n = 3) or C (ctxA/C, n = 4).

Consistent with the findings described in *Figure 2*, exposure to context A resulted in higher levels of LacZ induction in both CA1 and DG (*Figure 5B,D*; t-test, HC vs ctxA, in CA1, $t_9$ = 3.578, p<0.006; in DG, $t_9$ = 3.131, p<0.013). In ctxA/A mice, which were pre-exposed and re-exposed to the same context, the reactivation rates in CA1 were significantly higher than the activation rates, suggesting preferential activation of neurons that responded during pre-exposure by re-exposure; however, this preferential reactivation was not found in either HC or ctxA/C mice, whose experiences at pre-exposure and re-exposure were different (*Figure 5C*; ANOVA: group x activity rates interaction, $F_{2,1}$ = 13.99, p<0.0024; main effect of group, $F_{2,8}$ = 15.68, p<0.0017; main effect of activity rates, $F_{1,8}$ = 18.59, p<0.0026; Bonferroni post hoc test, for reactivation rate vs activation rate, p<0.001 in ctxA/A and p>0.05 in HC and ctxA/C; for reactivation rate, HC vs ctxA/A, p<0.001; ctxA/A vs ctxA/C, p<0.0001).

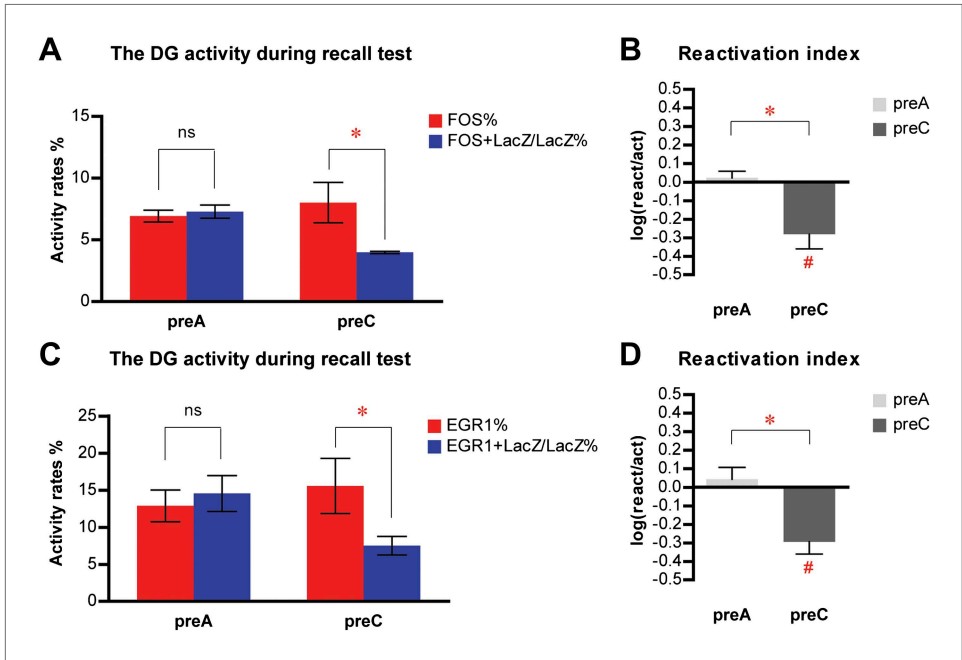

**Figure 4**. Similar results are obtained in the activity analysis of DGCs using either FOS or EGR1 as IEG markers in the same cohort of mice. (**A**) Activity analysis using FOS as IEG marker and RBFOX3 as neuronal marker demonstrates the selection of different populations of DGCs to represent different environmental inputs (ANOVA: group x activity rate interaction, $F_{1,1}$ = 9.038, p<0.017; Bonferroni post hoc test, reactivation rate vs activation rate, p>0.05 for preA mice, p<0.05 for preC mice; preA, n = 6; preC, n = 4). (**B**) Reactivation index calculated from the analysis using FOS as IEG marker. The index in preC is significantly smaller than preA (t-test, $t_8$ = 3.911, p<0.0045) and the chance level (one sample t-test, chance = 0, $t_3$ = 3.558, p<0.038), whereas the index in preA is not different from chance (one sample t-test, chance = 0, $t_5$ = 0.6153, p>0.56). (**C**) Activity analysis using EGR1 as IEG marker in the same cohort of mice has similar activity pattern as those analyzed by FOS (ANOVA: group x activity rate interaction, $F_{1,1}$ = 7.405, p<0.026; Bonferroni post hoc test, reactivation rate vs activation rate, p>0.05 for preA mice, p<0.05 for preC mice). The numbers of DGCs in the granule cell layers were quantified from DAPI images. (**D**) Reactivation index calculated from the analysis using EGR1 as IEG marker. The index in preC is significantly smaller than preA (t-test, $t_8$ = 3.519, p<0.0079) and the chance level (one sample t-test, chance = 0, $t_3$ = 4.403, p<0.022), whereas the index in preA is not different from chance (one sample t-test, chance = 0, $t_5$ = 0.6815, p>0.52). Asterisk indicates statistically significant difference between groups. Hash indicates statistically significant difference from chance. Data are shown in mean ± SEM.

In the DG, there was no preferential reactivation of neurons which were activated by pre-exposure in ctxA/A mice, whereas the reactivation rates were significantly lower than the corresponding activation rates in HC and ctxA/C mice (***Figure 5E***; ANOVA: group x activity rates interaction, $F_{2,1}$ = 12.98, p<0.0031; no main effect of group; main effect of activity rates, $F_{1,8}$ = 73.79, p<0.0001; Bonferroni post hoc test, for reactivation rate vs activation rate, p<0.001 in HC and ctxA/C and p>0.05 in ctxA/A; planned comparisons for reactivation rate, HC vs ctxA/A, p<0.05, ctxA/A vs ctxA/C, p=0.055). These results were further confirmed by the analysis of the reactivation index (***Figure 5F***; two-way ANOVA: group x region interaction, $F_{2,1}$ = 7.740, p<0.012; main effect of group, $F_{2,8}$ = 18.72, p<0.001; main effect of region, $F_{1,8}$ = 167.8, p<0.0001; compared to chance by one sample t-test: ctxA/A in CA1, $t_2$ = 7.241, p<0.0185; HC in DG, $t_3$ = 8.477, p<0.0034; ctxA/C in DG, $t_3$ = 6.909, p<0.0062). These data indicated that the pattern of neuronal activation and reactivation of HC mice was drastically different from that of ctxA/A mice, with reactivation rates in HC mice significantly lower than those of ctxA/A mice in both CA1 and the DG. Thus, HC activity does not seem to have a dramatic impact on the population reactivation pattern in CA1 and the DG. In addition, a similar activity pattern was found in ctxA/A and ctxA/C mice compared to that of preA and preC mice in the fear conditioning experiment (***Figure 3***), suggesting that the emotional value of contexts did not drastically influence the neuronal activity in CA1 or the DG of the hippocampus.

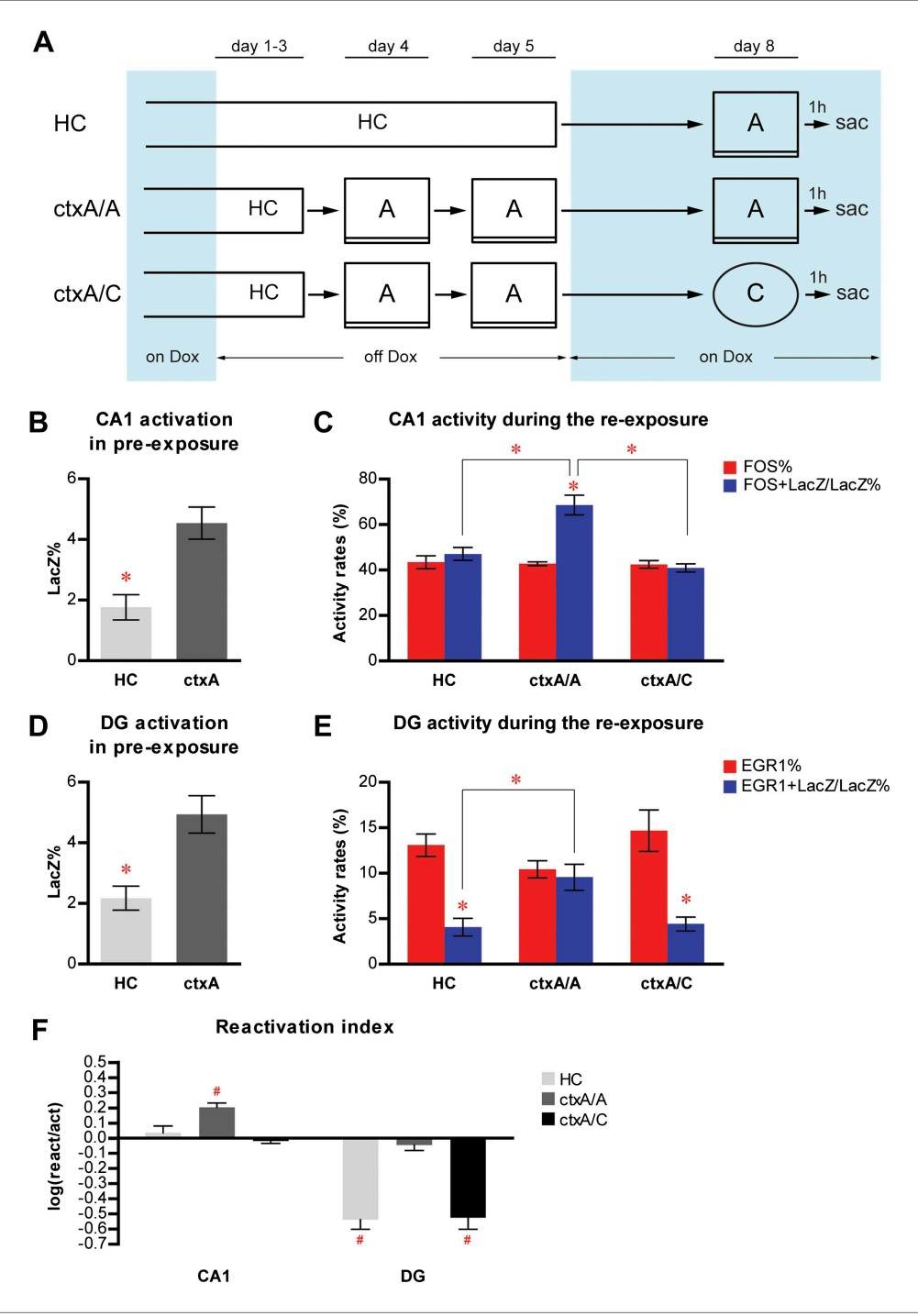

**Figure 5**. Neither home cage activity nor emotional value of context has a significant impact on reactivation patterns in CA1 and DG. (**A**) The experimental design. Dox treatment is illustrated by blue shading. (**B**) HC mice have a significantly lower number of LacZ-positive cells in CA1 compared to ctxA mice. (**C**) Preferential reactivation of CA1 neurons responding to pre-exposure by re-exposure occurs only in ctxA/A mice but not HC or ctxA/C mice. (**D**) HC mice have a significantly lower number of LacZ-positive cells in the DG compared to ctxA mice. (**E**) In HC and ctxA/C mice but not ctxA/A, the reactivation rate is significantly lower than the corresponding activation rate in the DG, suggesting different populations of DGCs are selected in response to distinct experiences (**F**) reactivation indexes analysis. Asterisk indicates statistically significant difference between groups or rates. Hash indicates statistically significant difference from chance. Data are shown as mean ± SEM (HC: home cage; sac: sacrifice).

## Selection of separate DGC populations to represent distinct events can be triggered by small changes in environmental inputs

Because the DG was postulated to function as a pattern separator to form distinct representations of similar inputs (*Marr, 1971*; *O'Reilly and McClelland, 1994*; *Rolls and Kesner, 2006*; *Rolls, 2010*), we asked whether small changes in contextual inputs might affect the selection of responding neuron populations in the DG. We trained a new cohort of mice for contextual fear conditioning in context A and subsequently tested them in either context A (testA, n = 10) or context B (testB, n = 11); the latter was modified from but still shared many common components with context A (similar but not the same) (*Figure 6A*, *Figure 2—figure supplement 1*, see 'Materials and methods'). testA mice displayed a higher level of freezing than testB mice (*Figure 6B*; t-test, $t_{19}$ = 2.123, p<0.047), suggesting that mice were able to detect the small changes in context.

We then examined neuronal activities and found that equivalent numbers of LacZ positive neurons were tagged in testA and testB mice in both CA1 and the DG (*Figure 6C,D*; t-test, CA1: $t_{19}$ = 0.2054, p>0.83; DG: $t_{19}$ = 1.319, p>0.20). ANOVA analysis of the activity rates in CA1 revealed that reactivation rates in both testA and testB mice were significantly higher than the corresponding activation rates, and there was no significant difference in either activation rates or reactivation rates between testA and testB mice (*Figure 6E*; ANOVA: main effect on activity rates, $F_{1,19}$ = 176.2, p<0.0001; no group effect, $F_{1,19}$ = 0.01493, p>0.90; no group x activity rate interaction, $F_{1,1}$ = 0.1676, p>0.68; Bonferroni post hoc test, activation rates vs reactivation rates, testA, p<0.0001, testB, p<0.0001; Bonferroni post hoc test, testA vs testB, activation rate, p>0.05, reactivation rate, p>0.05). Moreover, the reactivation indexes for CA1 were not significantly different between testA and testB mice (*Figure 6G*; t-test, $t_{19}$ = 0.4206, p>0.67) and were above chance in both groups of mice (*Figure 6G*; one sample t-test, chance = 0: testA, $t_9$ = 16.58, p<0.0001; testB, $t_{10}$ = 6.629, p<0.0001). These results extended our previous finding and indicated that recall-evoked preferential reactivation of CA1 neurons that were responsive during memory formation was resistant to perturbation by small alterations in environmental inputs.

In contrast to CA1, there was a significant interaction between group and activity rates in the DG (*Figure 6F*; ANOVA: group x activity rate interaction, $F_{1,1}$ = 36.94, p<0.0001, main effect on activity rates, $F_{1,19}$ = 42.19, p<0.0001; no group effect, $F_{1,19}$ = 0.8841, p>0.35). Similar to the results of the previous fear conditioning experiment (*Figure 3F*), the reactivation rate in the DG was significantly lower than the corresponding activation rates in testB but not testA mice (*Figure 6F*, Bonferroni post hoc test, activation rates vs reactivation rates, testA, p>0.05, testB, p<0.0001) and the reactivation rate, but not the activation rate, of the testB mice was significantly lower than that of testA mice (*Figure 6F*, Bonferroni post hoc test, testA vs testB, activation rate, p>0.05, reactivation rate, p<0.01). Moreover, the reactivation index in testB mice was significantly below chance (*Figure 6G*; one sample t-test, chance=0: $t_{10}$ = 8.321, p<0.0001) and was significantly lower than that in testA mice (*Figure 6G*; t-test, $t_{19}$ = 6.810, p<0.0001), which was not significantly different from chance (*Figure 6G*; one sample t-test, chance = 0: $t_9$ = 0.4784, p>0.64). These results demonstrate that small environmental changes were enough to evoke responses in distinct ensembles of DGCs but not CA1 neurons (*Figure 6G*), indicating that this selection of a unique population of DGCs to represent a particular event serves as a mechanism for the function of pattern separation.

## Discussion

By examining neuronal activity at the population level, we discovered that the DG and CA1 of the hippocampus displayed differential neuronal responses at a population level during learning and memory (see *Figure 7* for a model). In particular, our data revealed that the selection of separated populations of DGCs in the dorsal DG to represent similar but non-identical environmental inputs was a mechanism for pattern separation.

In the DG, distinct populations of DGCs that had limited overlaps were selected to represent two different events that were temporally separated (*Figure 7*). Moreover, the utilization of a separated DGC ensemble for encoding newly encountered events could be triggered by small changes in the environmental inputs (*Figures 6F,G and 7*). In contrast, CA1 network reactivation was not sensitive to the minor contextual alterations but could be affected by large input changes (*Figures 6F,G and 7*). The notion that different populations of DGCs are used to represent different inputs has also been suggested by a computation model based on data obtained by cellular compartment analysis of

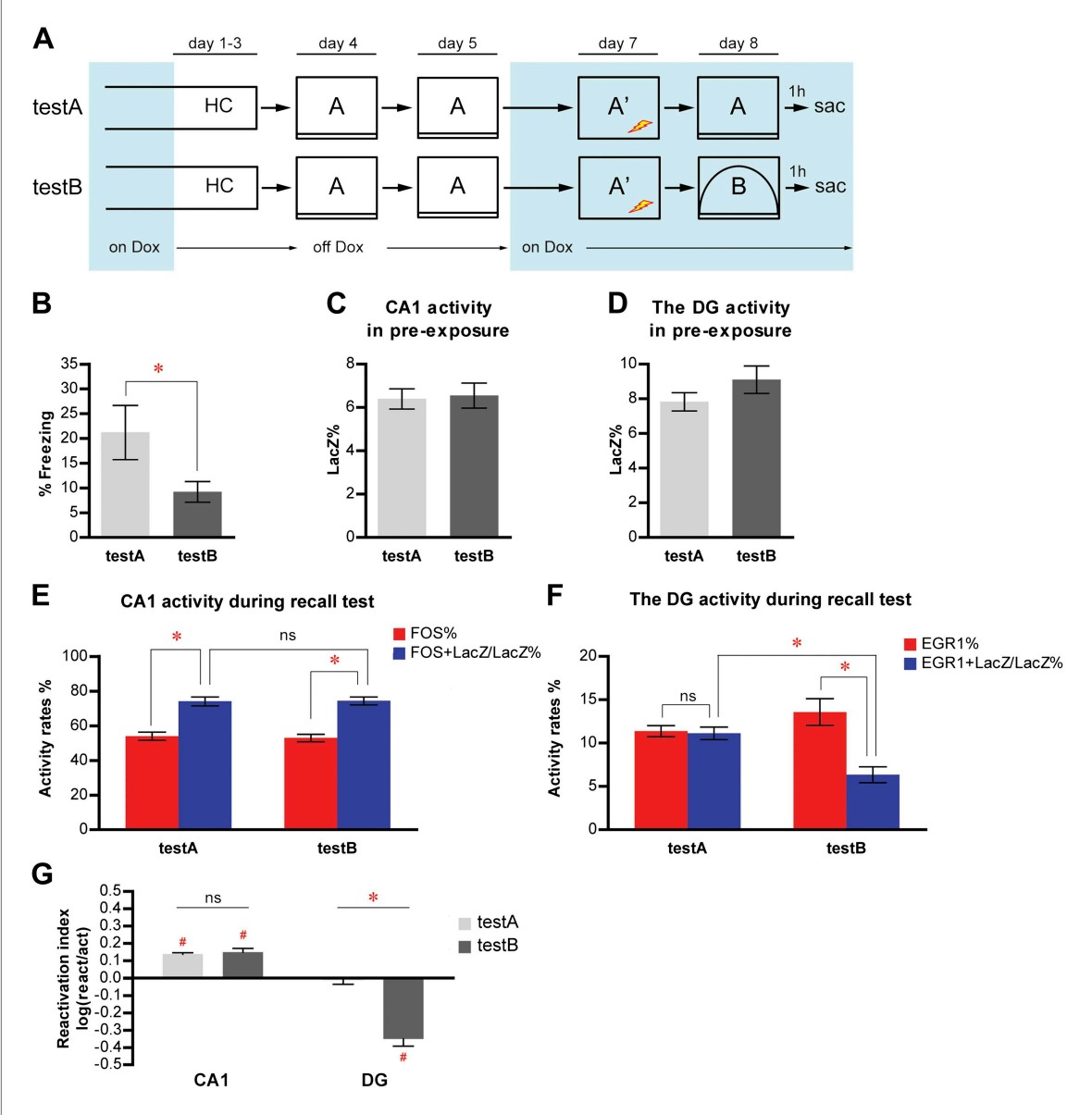

**Figure 6**. Population activities in the DG but not CA1 are sensitive to small changes in environmental inputs. (**A**) Mice subjected to the pre-exposure-immediate shock paradigm in context A were tested for memory retrieval in either context A (testA) or context B (testB), which was modified from context A. Dox treatment is illustrated by blue shading. (**B**) testA mice display significantly more freezing behavior than testB mice. (**C**) and (**D**) During pre-exposure, the percentage of LacZ-positive neurons in total population is not significantly different between testA and testB mice in either CA1 (**C**) or the DG (**D**). (**E**) Activity of CA1 neurons during retrieval test. While neither activation rates nor reactivation rates are significantly different between groups, reactivation rates are significantly higher than the activation rates in both testA and testB mice. (**F**) During the retrieval test, there is no preferential reactivation of LacZ-positive DGCs in testA mice, whereas LacZ-positive DGCs are significantly less likely to be reactivated in testB mice compared to testA mice. The reactivation rate is significantly lower than the corresponding activation rate in testB mice but not in testA mice. (**G**) Reactivation indexes suggesting the differential reactivations of learning-induced neuronal ensembles by recall in CA1 and the DG (ANOVA: region x group interaction, $F_{1,1} = 62.98$, p<0.0001; main region effect, $F_{1,19} = 215.4$, p<0.0001; main group effect, $F_{1,19} = 25.45$, p<0.0001). Asterisk indicates statistically significant difference between groups. Hash indicates statistically significant different from chance. Data are shown as mean ± SEM (ns: no significant difference; HC: home cage; sac: sacrifice).

temporal activity by fluorescence in situ hybridization (catFISH) of *Arc*, another IEG (*Chawla et al., 2005*). Consistent with our results, studies have shown that lesions in the DG but not in CA1 caused a deficit in discrimination of spatial locations of low but not high separations (*Gilbert et al., 2001*; *Goodrich-Hunsaker et al., 2008*). Similarly, blocking the plasticity in the DG resulted in a deficit in the

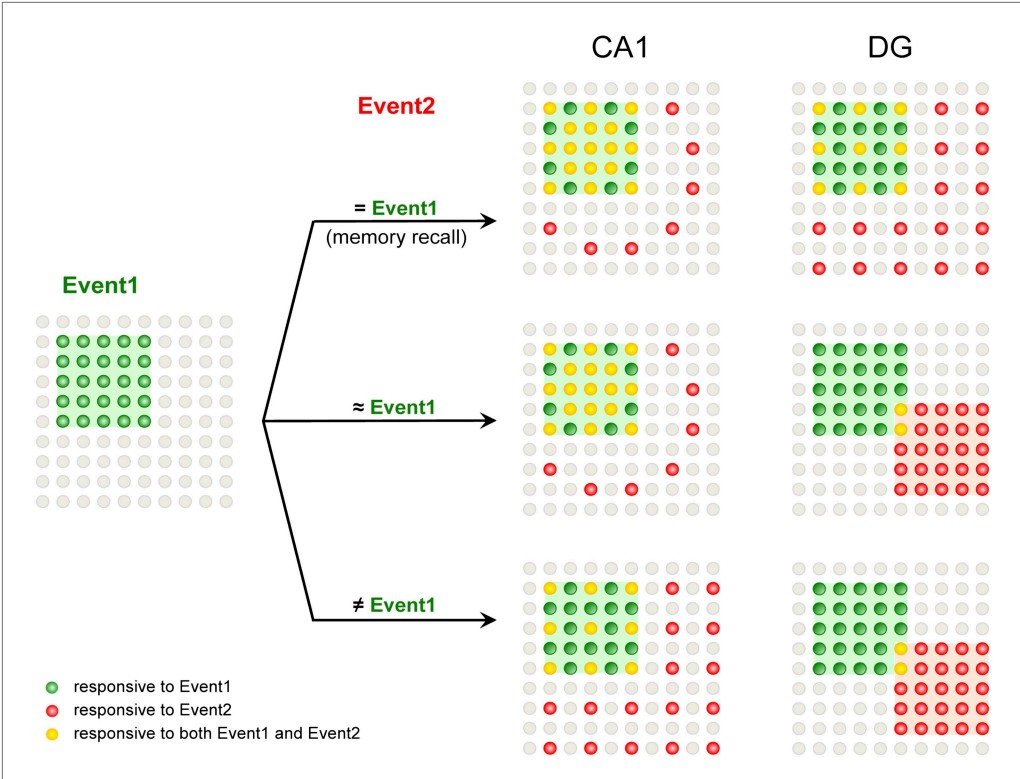

**Figure 7**. A model for population codes in CA1 and the DG during learning and memory. Experience and learning of an event (event 1, green) evoke activities in ensembles of neurons in CA1 and the DG (green cells). When mice subsequently encounter the same event, which will most likely induce memory recall (event 2 = event 1), the population of CA1 neurons responding to event 1 is preferentially reactivated (red cells), whereas DGCs responding to event 1 are reactivated at chance level (event 1-responsive DGCs have neither an advantage nor a disadvantage to be reactivated compared to the total DGC population). Neurons that are responsive to both events are in yellow. When mice encounter a second event that is similar but not identical to event 1 (event 2 ≈ event 1), there is still a preference to activate the CA1 neurons that are activated by event 1. However, in the DG, another population of DGCs that does not respond to event 1 will likely be selected to respond to event 2. Hence, small changes in inputs can evoke a population code change in the DG but not CA1, providing a neural basis for the pattern separation function of the DG. When mice encounter a second event that is drastically different from event 1 (event 2 ≠ event 1), CA1 neurons responding to event 1 are activated at chance level, whereas DGCs that did not respond to event 1 are selected to encode event 2.

discrimination of similar contexts (*McHugh et al., 2007*). Because DGCs are heavily innervated by local and hilar interneurons (*Houser, 2007*), inhibition of DGCs by these interneurons can be a potential neural mechanism underlying the population selection; future studies are needed to test this possibility.

Our findings seem to disagree with previous results of physiological studies showing that the same ensemble of DGCs was active in multiple different environments despite displaying distinct firing patterns (i.e., rate remapping but not global remapping; *Leutgeb et al., 2007*; *Alme et al., 2010*). One possible explanation for this discrepancy is that the physiological experiments and the experiments described here targeted different groups of neurons in the DG. Despite the fact that the identity of the neurons monitored by in vivo recording cannot be determined by simple histological analysis (*Neunuebel and Knierim, 2012*), it is postulated that the recorded neurons are likely to be newly born DGCs that are generated by adult neurogenesis (*Alme et al., 2010*; *Neunuebel and Knierim, 2012*), because the newborn DGCs are more excitable compared to their mature counterparts and are more likely to be recorded (*Deng et al., 2010*; *Aimone et al., 2011*). On the other hand, both mature and newly born DGCs were included in our analysis, with the mature DGCs representing the majority of the population (>90%) due to the low rate of adult neurogenesis (*Cameron and McKay, 2001*). In a preliminary effort to test this possibility, we measured the distances of LacZ-tagged and EGR1-positive

DGCs from the hilus and compared these distances to those of adult-born DGCs because adult-born DGCs tend to be located in the inner third of the granule cell layer (*Mathews et al., 2010*). While the majority of adult-born DGCs labeled by BrdU were located close to the hilus, the LacZ-positive and EGR1-positive DGCs were distributed throughout the granule cell layer and their locations were significantly further from the hilus compared to those of adult-born DGCs (*Figure 8*), suggesting that they represented a DGC population different from the adult-born DGCs. Future studies are needed to investigate whether responses of adult-born DGCs in learning and memory are different from those of their mature counterparts, even though it has been shown that the adult-born DGCs are important for spatial discrimination in mice (*Clelland et al., 2009*; *Creer et al., 2010*). In addition, the vast difference in kinetics between the in vivo recording studies and our study may also contribute to the inconsistency in the results. *Leutgeb et al. (2007)* studied the responses of DGCs to events that occurred minutes apart; however, there was a three-day interval between pre-exposure and re-exposure in our experiments. It is possible that the same group of neurons is recruited to encode for events occurring within a short time interval. Neurons that responded to one event had elevated levels of CREB1 for a short period of time, making them more likely to be recruited by another event occurring in this time window (*Silva et al., 2009*). Finally, although the expression of IEGs can reflect general activation of neurons, it remains unclear what physiological changes the expression of IEGs is corresponding to. It is possible for firing patterns to vary within the IEG positive population. Hence, our findings, together with data from physiological studies, suggest that the DG can carry out pattern separation through both global remapping and rate remapping.

Compared with the situation when the mice experience two different events, when mice encounter a previously experienced event for a second time (memory recall), there is an elevation in the reactivation level of the DGC population that was activated during the initial event learning. Although this level of reactivation did not rise above chance we detected a weak but significant correlation between the reactivation index and behavioral performance (*Figure 9*). This observation raises questions regarding which cortical-hippocampal pathway is reinstated by memory recall as well as whether reinstating the DG engram is sufficient and/or necessary for recall. A recent study showed that artificial reactivation of the DGCs involved in the acquisition of contextual fear conditioning was sufficient to induce the expression of fear memory in a neutral context (*Liu et al., 2012*), but the extent of reactivation adequate for memory recall remains unknown. Our findings suggest the possibility that a mild increase in the reactivation by releasing a DGC ensemble from suppression seems enough to trigger the successful memory retrieval and expression. On the other hand, the chance level of reactivation of a learning-induced DGC population by recall suggests an alternative possibility: that preferential reactivation of the DG may not be necessary for memory recall. Because of the existence of

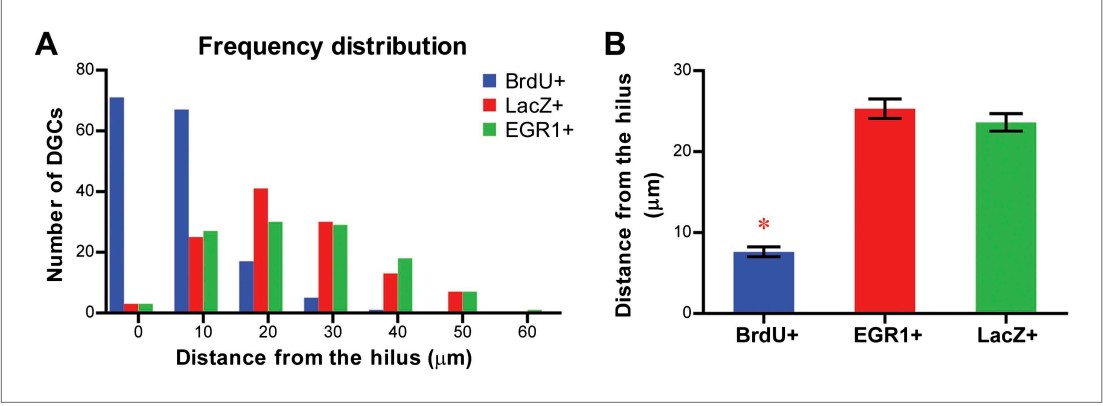

**Figure 8**. Comparison of the location of the LacZ-positive and EGR1-positive DGCs with that of adult-born granule cells in the granule cell layer of the DG. The distance of each cell from the hilus was measured using Metamorph. Adult-born granule cells were labeled by treating mice with water containing BrdU for one week. Treated mice were perfused more than 6 weeks later for histological examination of the locations of BrdU-labeled cells in the granule cell layer. (**A**) Frequency distribution showing that the majority of the BrdU-positive cells are located close to the hilus, whereas both LacZ-positive and EGR1-positive populations were distributed across the granule cell layer. (**B**) The distance from the hilus is significantly shorter in BrdU-positive cells compared to that of the EGR1-positive or LacZ-positive cells (ANOVA: $F_{2,392}$ = 120.6, $p<0.0001$; Bonferroni post hoc test, $p<0.001$).

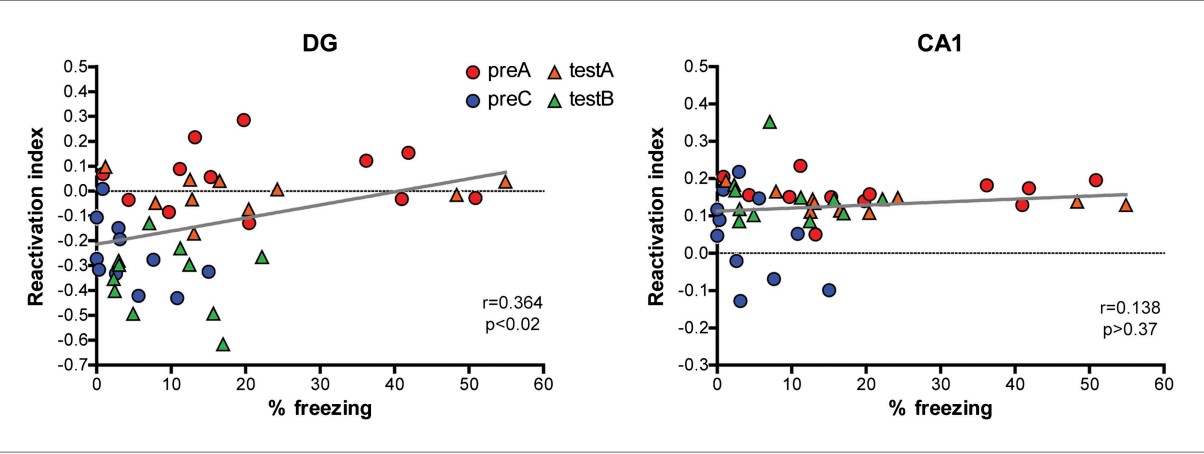

**Figure 9**. Correlations between reactivation indexes and behavioral performance in the contextual fear conditioning experiments. A weak but significant correlation between behavioral performance and the reactivation index was detected in the DG (r = 0.3641, p<0.016) and no significant correlation was found in CA1 (r = 0.1379, p>0.37).

multiple parallel pathways between the cortex and hippocampus, it is conceivable that memory retrieval may not necessarily rely on the EC→DG→CA3 pathway. Although this hypothesis remains to be tested directly, several lines of evidence support it. First, lesion of the DG affects encoding but not retrieval of spatial information, as indicated by behavioral studies (*Lassalle et al., 2000*; *Lee and Kesner, 2004*). Moreover, memory retrieval with the full set of recall cues is not affected by blocking the transmission between the DG and CA3 (*Nakashiba et al., 2012*). Finally, computational studies also suggest that, while the DG inputs to CA3 may be critical during learning, retrieval of memory may rely on direct pathways from EC to CA3 (*Treves and Rolls, 1992*; *Rolls, 2010*). According to this theory, CA3 neurons involved in memory encoding are expected to be preferentially reactivated during memory retrieval. Unfortunately, we were unable to test this hypothesis in the current study due to technical limitations. However, we have demonstrated that CA1 neurons involved in encoding were preferentially reactivated by memory recall. In summary, our results suggest that, in a complex neural network, successful memory recall may not preferentially reactivate all the responsive pathways that are involved in memory formation.

In contrast to the DG, CA1 neurons activated during contextual memory formation were preferentially reactivated upon retrieval of the memorized contextual information, even though the context was later associated with an emotional value (in our case, fear). This observation is in agreement with the notion that the hippocampus can automatically encode ongoing events, whereas the association of these events with an emotional value occurs in other brain structures, such as amygdala (*Rudy and O'Reilly, 2001*; *Stote and Fanselow, 2004*). Indeed, similar reactivations of CA1 neurons were observed when animals were re-exposed to a previously experienced environment without a change in emotional value (*Figure 5*, ctxA/A group). Similar to our findings, equivalent levels of IEG induction were found in the hippocampus by subjecting mice to contextual fear conditioning training (context exposure plus foot shocks) or by exposing mice to the context without foot shocks (*Ramamoorthi et al., 2011*; *Liu et al., 2012*). In addition, the role of the amygdala in the association of events with an emotional value is supported by the findings that neurons in the basal lateral amygdala activated during fear conditioning training are preferentially reactivated by the retrieval of contextual fear memory (*Reijmers et al., 2007*) and that post-learning elimination of amygdala neurons involved in fear learning erases the fear memory (*Han et al., 2009*). Unlike in the DG, we were not able to detect a linear correlation between the CA1 reactivation and freezing behavior of mice under our experimental conditions (*Figure 9*). Particularly, in the experiment involving only small contextual changes, the reactivation index in CA1 did not change accordingly, although the alteration in environmental inputs seemed to be detected by mice, as reflected by their freezing behaviors. It is possible that the reactivation of CA1 may be necessary but not sufficient to drive the behavior under certain circumstance (e.g. when the input difference is detected by the DG/CA3 network). Given that remapping in CA1 is less sensitive to

changes in environmental cues compared to CA3 and the DG (*Leutgeb et al., 2004*, *2007*), the small alteration in our experiment may not be significant enough to trigger a global remapping of CA1 neurons, although it remains possible that the firing patterns of activated neurons may be different. In addition, behavioral studies have shown that CA1 is dispensable for spatial pattern separation (*Gilbert et al., 2001*). In summary, our findings are not only consistent with previous reports that CA1 is critically involved in both encoding and retrieval of spatial and contextual information but also suggest that the same CA1 ensemble used for memory formation is likely to be reactivated by recall of the same memory trace.

## Materials and methods

### Animals and treatments

The TetTag transgenic mice were obtained from Mark Mayford's lab and re-derived into a mixed background of C57BL/6 and balb/c. The mice were bred by intercrossing the hemizygous *Fos-tTA:shEGFP* line with the hemizygous *tetO-tTA\*:tau-lacZ* line. All mice had food and water ad libitum. The breeding pair and newborn pups were treated with water containing 10 μg/ml dox and 1% sucrose. After weaning, the double transgenic TetTag mice were raised on a 40 mg/kg dox diet. Mice were at least 11 weeks old at the start of the experiments and were group housed until 1 week before the experiments. For BrdU labeling, mice were treated with water containing 2 mg/ml BrdU and 2% sucrose for 1 week. The mice were euthanized >6 weeks later to examine the location of BrdU labeled DGCs. All experimental procedures were approved by the Institutional Animal Care and Use Committee at The Salk Institute for Biological Studies.

### Behavioral procedures

#### The enriched environment experiment

Mice were individually housed 1 week before the experiment. While some mice were maintained on the dox diet, others were removed from the dox treatment by replacing the dox diet with regular mouse chow for 3 days. On the fourth day, both groups of mice were placed in an enriched environment in a transparent plexiglass box measured 36 inches (L) × 36 inches (W) × 12 inches (H) and containing two running wheels, three plastic huts and several plastic tunnels. After 3 hr, the mice were removed from the enriched environment and were immediately sacrificed.

#### The experiment comparing context A vs home cage

In this experiment, mice were individually housed 1 week before the experiment and were removed from dox treatment and remained undisturbed in their HC for 3 days. On days 4 and 5, some of the mice were exposed to a contextual fear conditioning chamber (context A in *Figure 2—figure supplement 1*) for 10 min each day; the others remained in their HC. After contextual exposure on day 5, all mice were treated with a 1 g/kg dox diet until being sacrificed 3 days later.

#### Contextual fear conditioning: experiment 1

The fear conditioning apparatus and software were obtained from Med Associates, Inc (St. Albans, VT). We used a protocol that combined immediate shock with contextual pre-exposure to train mice for the contextual fear conditioning (*Fanselow, 1990*). We chose this protocol because the context learning phase can be well separated temporally from the memory recall phase to suit the slow kinetics in the TetTag system (*Reijmers et al., 2007*). In addition, identical environmental inputs could be delivered at the pre-exposure and re-exposure. Although we did not intentionally design our paradigm for the behavioral readout, we did observe differential behavioral responses under different experimental conditions (*Figures 3B and 6B*). TetTag mice were individually housed 1 week before the experiment and were handled 3 min per day for 3 to 5 days. On day 1 of the experiment, mice were removed from dox treatment and were undisturbed in their HC until pre-exposure on day 4. On days 4 and 5, mice were divided into two groups and were subjected to pre-exposure for 10 min on each day. One group of mice was exposed to the conditioning chambers in sound attenuated boxes (context A) and the other group was exposed to context C (*Figure 2—figure supplement 1*). To prevent the generalization of fear response (*McHugh et al., 2007*), the wired grid, from which foot shocks were delivered, was covered by a plastic board in context A. Context C, completely different from context A, is located in another testing room, is modified from an open field chamber by inserting a dark box made of

plexiglass and is scented with vanilla extract. The passage between the dark and light compartments is blocked, restricting the mouse within the light compartment. Subsequent to the pre-exposure procedure on day 5, all mice were put on a 1 g/kg dox diet to prevent the further tagging of activated neurons. On day 7, both groups of mice were subjected to the immediate shock protocol in context A', which was identical to context A except that the plastic floor was removed to allow the eliciting of shock through grid wires (*Figure 2—figure supplement 1*). The shock (0.7 mA, 2 s) was delivered 5 s after mice were placed in the chamber. 24 hr later, mice were re-introduced to context A for 3 min and were returned to their HC after the test. Although we refer to this test as the recall test, it should be noted that, during this re-exposure, mice could retrieve the original contextual memory (formed during pre-exposure) and/or encode the newly encountered context depending on their previous experiences. Behaviors of the mice were recorded and analyzed using video freeze software (Med Associates). Mice were euthanized 1 hr after the recall test and their brains were dissected out for analysis.

## Contextual fear conditioning: experiment 2

The behavioral procedure in experiment 2 was very similar to that in experiment 1. Instead of dividing the mice into two groups during pre-exposure, all mice were pre-exposed to context A and subjected to immediate shock in context A' 2 days later. The mice were divided into two groups during memory recall to test for contextual discrimination. One group of mice was returned to context A for 3 min and the other group of mice was placed in context B for 3 min. Context B was modified from context A by altering the shape of the chamber with a curved plastic board, changing the olfactory cues, and changing distal visual cues with posters on the walls of the sound attenuating box (*Figure 2—figure supplement 1*). Mice were euthanized 1 hr after the recall test and their brains were dissected out for analysis.

## Sequential contextual exposure experiment

The behavioral procedure in this experiment was very similar to that in experiment 1. While one group of mice was pre-exposed to context A, the other group of the mice was kept in their HC during the dox-off window. 3 days after the mice were put back on dox treatment, the latter group was exposed to context A before euthanasia. The group pre-exposed to context A was divided into two sub-groups, with one sub-group re-exposed to context A and the other sub-group exposed to context C. All mice were euthanized 1 hr after the second context exposure and their brains were dissected out for analysis.

## Histology and immunohistochemistry

Mice were sacrificed and brain sections were prepared according to previously reported procedures. Briefly, 1 hr after contextual re-exposure, mice were anesthetized with ketamine (100 mg/kg) and xylazine (10 mg/kg) and were perfused transcardially with saline followed by 4% paraformaldehyde in PBS. The brains of mice were dissected out and post-fixed with 4% paraformaldehyde overnight at 4°C and equilibrated with 30% sucrose. Coronal sections of 40 μm were cut throughout the hippocampal region and stored in the tissue preservation solution at −20°C. Brain sections from a one-in-twelve series were selected for immunostaining. The sections were either double stained with anti-EGR1 and anti-LacZ antibodies or triple stained with anti-FOS, anti-LacZ and anti-RBFOX3 (aka NeuN) antibodies. The following primary antibodies were used: mouse anti-LacZ (1:10,000; Promega, Madison, WI/Fisher, Pittsburgh, PA), goat anti-LacZ (1:1000; Serotec/Biogenesis, Raleigh, NC), rabbit anti-FOS (1:800; Santa Cruz, Dallas, TX), rabbit anti-EGR1 (1:800; Santa Cruz) and mouse anti-RBFOX3 (1:100; Millipore, Billerica, MA), rat anti-BrdU (1:500; Accurate, Westbury, NY). All secondary antibodies were used in 1:250 dilutions and were from Jackson ImmunoResearch. To visualize cell nuclei, all sections were stained with DAPI (0.5 μg/ml).

## Confocal microscopy and image quantification

Confocal images were acquired by either a Bio-Rad confocal microscope or a Zeiss LSM 710/780 laser scanning confocal microscope. Images showing the overview of the hippocampus in *Figure 1—figure supplement 2* were collected on one z focal plane using a 25× lens with 8 × 4 tiling. Images showing the overview of CA1 and the DG in *Figure 2* were collected by a 25× lens with 4 × 2 tiling. For all other images, Z-series (10–20 μm for CA1 and 20 μm for the DG) with a 2-μm interval were acquired using a 40× lens. Images illustrating CA3 in *Figure 1—figure supplement 2* were obtained using 2 × 3 tiling. All images used for quantification in the fear conditioning experiments that collected on Zeiss LSM

scopes were acquired using 2 × 1 tiling except for those used for DG quantification in *Figure 6* (no tiling). Typically, four to five images were analyzed for each animal in each region. The experimenter was blind to the behavioral history of the mice for all quantifications.

## Quantification in CA1

Quantification was performed on one of the focal planes in the z-series. FOS staining was used for quantification in CA1 due to the relative low percentage (~50%) of labeling of this IEG marker. Four types of cells were quantified in CA1 pyramidal layer in each image: RBFOX3 positive or DAPI positive cells, FOS-positive cells, LacZ-positive cells, and FOS+LacZ double-positive cells; the latter three populations were also positive for DAPI or RBFOX3. Each type of cell was counted using the 'manually count objects' function of the Metamorph software. For double-positive cells, adjacent pictures in the z-series were used to validate the co-localization. The summations of cell counts in each category were obtained from four to five images along the rostral-caudal axis of the dorsal hippocampus. From these quantifications, we calculated the activation rate and reactivation rate according to the following formulas:

$$activation\ rate = \frac{the\ number\ of\ FOS + cells}{the\ number\ of\ RBFOX3 + cells},$$

$$reactivation\ rate = \frac{the\ number\ of\ FOS + LacZ + cells}{the\ number\ of\ LacZ + cells}.$$

## Quantification in the DG

EGR1 staining was used for quantification in the DG due to the relatively high percentage (~10–15%) of labeling of this IEG marker. In a subset of mice, the results were also confirmed by using FOS as the IEG marker and RBFOX3 as the marker for all neurons (*Figure 4*). Quantification in the DG was carried out by two methods to analyze the data in contextual fear conditioning experiment 1 (*Figure 3* and *Figure 3—figure supplement 2*). In the first method, four types of cells were quantified in the granule cell layer on a single focal plane of each image: DAPI-positive cells, EGR1-positive cells, LacZ-positive cells, and EGR1+LacZ double-positive cells. Cells in the latter three categories were also positive for DAPI. Similar to that of CA1, the activation rate and reactivation rate according to the following formulas:

$$activation\ rate = \frac{the\ number\ of\ EGR1 + cells}{the\ number\ of\ DAPI + cells},$$

$$reactivation\ rate = \frac{the\ number\ of\ EGR1 + LacZ + cells}{the\ number\ of\ LacZ + cells}.$$

In the second method, quantification was performed on the entire z-series of confocal images using Metamorph software. Three types of cells were quantified manually by examination of each focal plane in the z-series: EGR1-positive cells, LacZ-positive cells, and EGR1+LacZ double-positive cells. For double-positive cells, adjacent planes in the z-series were used to validate the co-localization. The total number of DGCs in the z-series was calculated according to the following formula:

$$total\ number\ of\ DGCs\ in\ images = DG\ area \times depth \times DGC\ density,$$

in which DG area (in μm²) was measured from a DAPI image resulting from maximum projection of z-series by tracing the outline of the dentate granule cell layer. Constant values were used for the depth and DGC density. The depth was 20 μm, which was scale of the z-series. We used 1.1/1000 μm³ as the DGC density (see below). Similar to CA1, the activation rate and the reactivation rate were calculated according to the following formulas:

$$activation\ rate = \frac{the\ number\ of\ EGR1 + cells}{total\ number\ of\ DGCs\ in\ images},$$

$$reactivation\ rate = \frac{the\ number\ of\ EGR1 + LacZ + cells}{the\ number\ of\ LacZ + cells}.$$

Given that estimation was applied in the second method, the resulted data might not be absolutely accurate. However, the between-group comparison should still be valid because the same estimation was applied for both experimental and control mice. Furthermore, data resulting from the second quantification approach were similar to those of the first approach (*Figure 3* and *Figure 3—figure supplement 2*). Therefore, the second quantification method was used for analyzing data in the subsequent experiments.

### The reactivation index

In both CA1 and the DG, given that the reactivation rate was directly influenced by the corresponding activation rate in each mouse, we normalized the data using the following formula:

$$reactivation\ index = log_{10}\left(\frac{reactivation\ rate}{activation\ rate}\right).$$

The reactivation index provides a linearized measure of the degree of recall-induced reactivation and is used to compare the reactivation between animal groups and between different regions of the hippocampus.

### Measurement of DGC density

Previous reports have shown that the density of DGCs in C57BL/6 mice is $1.10 \pm 0.04/1000\ \mu m^3$ (*Kempermann et al., 1998*). We also measured the density of DGCs in TetTag mice. Brain sections (one-in-twelve series) from five mice were stained with anti-NeuN antibody. Three confocal z-series were taken from each section, with one each from the following DG areas: suprapyramidal blade, infrapyramidal blade and the vertex region. The z-series was taken at the thickness of 10 μm (with an interval of 1 μm) because the average diameter (width) of a DGC is about 10 μm. The number of DGCs was counted on the images resulting from maximum projection of z-series and the area of the dentate granule layer was outlined and measured using Metamorph. The volume of the granule cell layer in the image was the product of the image depth (10 μm) and the area of granule cell layer. The DGC density calculated from this measurement was $1.14 \pm 0.02/1000\ \mu m^3$. Because this value is very close to the reported DGC density by *Kempermann et al. (1998)*, we used $1.1/1000\ \mu m^3$ as the density to estimate the total number of DGCs in our quantification (see above). The total numbers of neurons quantified in each region for the fear conditioning experiments are shown in *Table 1*.

## Statistics

All statistical analyses were performed using Prism Graphpad software. Data were analyzed with unpaired t-test, one-way ANOVA, two-way ANOVA with repeated measures followed by post hoc Bonferroni tests as indicated. Comparison with chance level was done using one sample t-test using 0 as a theoretical mean. The relationship between the reactivation index and behavioral performance was measured by simple linear correlations (Pearson correlation). All data were presented as mean ± SEM.

**Table 1.** Total numbers of neurons evaluated in experiments

| | CFC experiment 1 | | CFC experiment 2 | |
|---|---|---|---|---|
| | preA | preC | testA | testB |
| n | 12 | 11 | 10 | 11 |
| CA1 | 486 ± 43 | 533 ± 26 | 544 ± 27 | 573 ± 20 |
| DG* | 4492 ± 222 | 4267 ± 345 | 2575 ± 177 | 2590 ± 102 |

*Calculated number. CFC: contextual fear conditioning.

## Acknowledgements

We would like to thank Drs Yan Li, James B Aimone, June Yao, Chunmei Zhao, Tiago Goncavas, Yangling Mu, and Jinju Han for discussions and comments and Ms. Mary Lynn Gage for editorial comments on the manuscript. Images were captured using confocal microscopes at Waitt advanced biophotonics center core facility at the Salk Institute.

## Additional information

### Funding

| Funder | Grant reference number | Author |
| --- | --- | --- |
| James S. McDonnell Foundation | | Fred H Gage |
| Lookout Fund | | Fred H Gage |
| Kavli Institute for Brain and Mind | | Fred H Gage |
| National Institutes of Health | MH-090258, NS-050217, AG-020938 | Fred H Gage |

The funders had no role in study design, data collection and interpretation, or the decision to submit the work for publication.

### Author contributions

WD, Conception and design, Acquisition of data, Analysis and interpretation of data, Drafting or revising the article; MM, Analysis and interpretation of data, Revising the article, Contributed unpublished essential data or reagents; FHG, Conception and design, Drafting or revising the article

### Ethics

Animal experimentation: All experimental procedures were approved by the Institutional Animal Care and Use Committee at The Salk Institute for Biological Studies (protocol#09-060).

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
