## [Decision Letter]

Thank you for choosing to send your work entitled “Selection of distinct populations of dentate granule cells in response to inputs as a mechanism for pattern separation” for consideration at *eLife*. Your article has been favorably evaluated by a Senior editor and 3 reviewers, one of whom is a member of our Board of Reviewing Editors.

The Reviewing editor and the other reviewers discussed their comments before we reached this decision, and the Reviewing editor has assembled the following comments based on the reviewers' reports. General assessment and substantive concerns to be addressed during revision:

1. The authors use fear conditioning and active ensemble labeling (TetTag mouse and IEG IHC) in the CA1 and DG of the dorsal hippocampus to study how memory recall engages previously active neurons across these subregions. Their main findings are that recall leads to “reactivation” in CA1, but not in the DG. However, when faced with a novel context the DG actually remapped to a greater extent than would be expected by chance, which the authors suggest as a mechanism of pattern separation. First, the equivalent labeling observed in CA1 and the DG in the TetTag mice was surprising; as is obvious from this paper, this is not the case for endogenous FOS staining, which typically shows a 10x increase in CA1 compared to DG. Although their scoring tries to control for this, the authors should discuss how this artifact of the TG line could skew their results. In a similar vein, although they note the lack of FOS/lacZ overlap in Figure 1C, this should be quantified and noted in the table.

2. As the authors note, these data are in partial conflict with some earlier published results. First, physiology (Leutgeb et al 2007) has previously suggested no “global” remapping in the DG, which the authors suggest may be due to a bias towards sampling adults born GCs during recording. This is an interesting suggestion. As the authors should be able to partition their current dataset – a simple method could be distance from the hilus – to test if this bias is present in their study, it is not clear why it was not included. In addition, they raise the recent Liu et al (2012) study that demonstrated memory “reactivation” in the DG could drive fear behavior in a neutral context. It would strengthen the relationship between the current work and that paper if the authors could correlate the reactivation score and the freezing behavior of the individual mice to see if a correlation can be observed.

3. One surprising observation that is unaddressed in the discussion: in experiment 1, high freezing in the preA mice paralleled the increase in CA1 reactivation index, while low freezing in the preC mice paralleled a decrease in the DG reactivation index. In experiment 2, the behavioral result is similar: high freezing in A, low freezing in B. However, the CA1 reactivation index is identical in both conditions. This raises the question: why is there low freezing (on average) in B while there is equivalent reactivation in CA1? Is the CA1 ensemble independent of the behavior? This should be discussed in detail.

4. The main conclusions of the paper rely on comparisons between 2 populations of activated neurons, using measures that are very different from each other. lacZ expression is used to measure long-term activation of neurons (i.e., in response to the first experience, days ago) whereas c-fos or zif268 is used to measure recent activity (for the 2nd experience). In CA1 lacZ is expressed in ∼6% of neurons whereas c-fos is expressed in ∼50% of neurons, nearly a 10x difference. In the DG zif268 is expressed at about twice the lacZ levels (12% vs 6%). The dramatic differences in activity-dependent expression of these two markers make it difficult to understand how they can be compared. In other words, it seems that they don't represent activity equally at all. Granted, this problem is somewhat tempered by the fact that activation and reactivation measurements are both calculated as % of cells expressing IEGs zif or fos (% of the total population vs % of the lacZ population). Clearly these two populations of cells are differentially activated in some cases and maybe it's the case that lacZ expression does not identify a qualitatively different type of activity, but just identifies fewer activated cells (e.g., because it has a higher threshold?). It would be more convincing if this was demonstrated. Right now there are factors that make one wonder if they might just be detecting different types of activity.

5. Due to the slow onset of the tTA-tetO system, the authors remove Dox 3d before subjecting mice to experiences for activating neuronal ensembles. Thus, there is a significant amount of time (and therefore experience) during which DG neurons could be activated (and express lacZ) based on home cage experiences. It is shown that home cage lacZ expression is lower than context-induced activation but how do we know that, in the dual context experiments, lacZ expression is due to the brief context pre-exposure rather than home cage activity? This could be addressed by showing that there is very little IEG expression in lacZ cells when the mice are taken off Dox but not exposed to the first context. A related issue lies with presentation of the experimental design. In the figures (e.g., Figure 1b, Figure 3a, and Figure 5a) the “on Dox” arrows extend significantly into the portion of the experimental timeline where the animals are not on Doxycicline. This gives the impression that Dox is removed at the time of the 2nd experience when in fact it is removed much earlier, in the home cage situation.

6. It is not clear why the immediate shock contextual fear paradigm is used. Most previous studies that have examined the role of the DG in pattern separation have looked at activity in response to exploration of environments that are not associated with reward or shock. This doesn't necessarily have to be the case but for the current study the additional experiences associated with the learning process seem to confound interpretations. For example, to what extent does the 2nd experience activate neurons due to emotional content (recalling a memory of being shocked) vs. due to the fact that it's simply a different spatial environment? Also, why was the shock floor covered during pre-exposure and test but not during the shock? Why cover it at all? Thus, even animals that were exposed to the same context for both experiences (in order to see if the same population of DG neurons encodes re-experience of the same context) were also exposed to an additional similar context (one without the covered floor). So, what memory are the animals really recalling? Everything would be much cleaner if the mice were just exposed to one context and then a second context, with no intervening shocks or altered environments.

7. The cell quantification methods are poorly described. First, what is meant by “total number of DGCs (dentate granule cells)” and why was it, along with density, calculated? Some mention of calculations is made but not described so it may become clear if described better. Perhaps, since the proportion of DG neurons that are activated is quite low, the authors manually counted all IEG and lacZ expressing cells but did not want to manually count all the unactivated cells (NeuN and DAPI cells). So there was some calculation of local DG neuron density, which was then used to extrapolate to estimate the actual number of DG neurons that was sampled. Is this correct? In any case, there is some concern about these methods since this density-to-total-number-of-DG-neurons conversion can introduce biases. This is significant because the activation rate is based on this number (IEG/total number of DG neurons) whereas the reactivation rate is not (both numerator and denominator are based on manual cell counts with no density conversions) and many of the key comparisons are between activation and reactivation rates. CA1 analyses are also based on manual cell counts so comparisons there are also problematic.

---

## [Author Response]

*1. The authors use fear conditioning and active ensemble labeling (TetTag mouse and IEG IHC) in the CA1 and DG of the dorsal hippocampus to study how memory recall engages previously active neurons across these subregions. Their main findings are that recall leads to “reactivation” in CA1, but not in the DG. However, when faced with a novel context the DG actually remapped to a greater extent than would be expected by chance, which the authors suggest as a mechanism of pattern separation. First, the equivalent labeling observed in CA1 and the DG in the TetTag mice was surprising; as is obvious from this paper, this is not the case for endogenous FOS staining, which typically shows a 10x increase in CA1 compared to DG. Although their scoring tries to control for this, the authors should discuss how this artifact of the TG line could skew their results. In a similar vein, although they note the lack of FOS/lacZ overlap in Figure 1C, this should be quantified and noted in the table*.

We agree with the reviewers that the rate of tagging is low compared to the endogenous IEG labeling and we believe that this problem is due to inherit shortcomings of the transgenic approach – the efficiency of expression of transgenes is less than 100% and varies among brain regions. In the same TetTag mice, the low efficiency of LacZ induction has also been observed in basolateral amygdala in the original study by Reijmers et al. (Reijmers et al., 2007). To test the representativeness of the tagged population, we measured the intensity of FOS staining in the FOS+LacZ+ neurons and FOS+LacZ- neurons in mice that were perfused after an enriched environmental exposure during the dox-off window. In these mice, expressions of LacZ and FOS should theoretically be induced in the same population of neurons. We found equivalent intensity between the FOS+LacZ+ and FOS+LacZ- populations, suggesting that tagged neurons likely to be representative of the overall activated population. The data are presented in Figure 1–figure supplement 3b-e. However, we cannot formally rule out the possibility that the only specific population (e.g., neurons with highest activities) of activated neurons is tagged. In this scenario, our conclusions will only apply to this specialized population, though we think this possibility is unlikely. We discuss this issue in the revised manuscript. Contrary to the lack of cFos and lacZ overlap, we have noticed that “many lacZ-positive cells also co-expressed cFos” and, as suggested by the reviewers, we quantified the data and tabulated the results in Figure 1–figure supplement 3a.

*2. As the authors note, these data are in partial conflict with some earlier published results. First, physiology (Leutgeb et al 2007) has previously suggested no “global” remapping in the DG, which the authors suggest may be due to a bias towards sampling adults born GCs during recording. This is an interesting suggestion. As the authors should be able to partition their current dataset – a simple method could be distance from the hilus – to test if this bias is present in their study, it is not clear why it was not included. In addition, they raise the recent Liu et al (2012) study that demonstrated memory “reactivation” in the DG could drive fear behavior in a neutral context. It would strengthen the relationship between the current work and that paper if the authors could correlate the reactivation score and the freezing behavior of the individual mice to see if a correlation can be observed*.

The possibility that those DG neurons responding to different environmental input by rate remapping are likely to be adult-born DGCs in the study of Leutgeb et al (2007) is raised not only by us but also by the authors in a follow up study (Alme et al., 2010) and by Knierim and colleagues in a recent study (Neunuebel and Knierim, 2012). As pointed out by the reviewers, the adult-born DGCs are likely located in the inner granule cell layer close to the hilus (Mathews et al., 2010). Therefore, it is possible to test whether the populations examined are different between the studies by determining the location of neurons. While it is difficult to determine the locations of the recorded neurons in physiological studies (Neunuebel and Knierim, 2012), we found that both LacZ+ and EGR1+ populations were distributed across the granule cells layer, significantly different from the distribution pattern of the BrdU labeled newborn DGC population. These new results are presented in Figure 8 and discussed accordingly. However, it remains possible that some newborn DGCs are included in our study and future studies are required to address the question as to whether the response pattern of the newborn DGC population is different from that of the mature DGC population in learning and memory.

Prompted by the reviewers, we have now included the result of a correlation analysis between the reactivation index and the freezing behavior of the individual mice in Figure 9. We found that the reactivation index in the DG was weakly correlated with the behavioral performance, whereas no linear correlation was found between the CA1 reactivation index and behavior. This observation is consistent with the postulation that the DG is more engaged in pattern separation compared to CA1, given that our behavioral protocol was designed for testing animals' ability in discriminating different environmental inputs.

*3. One surprising observation that is unaddressed in the discussion: in experiment 1, high freezing in the preA mice paralleled the increase in CA1 reactivation index, while low freezing in the preC mice paralleled a decrease in the DG reactivation index. In experiment 2, the behavioral result is similar: high freezing in A, low freezing in B. However, the CA1 reactivation index is identical in both conditions. This raises the question: why is there low freezing (on average) in B while there is equivalent reactivation in CA1? Is the CA1 ensemble independent of the behavior? This should be discussed in detail*.

We discuss the involvement of CA1 in behavioral performance in the revised manuscript as suggested by reviewers. We find that the CA1 reactivation index is not linearly correlated with freezing behavior of mice under our experimental conditions (Figure 9). Particularly, in the contextual fear conditioning experiment 2, involving only small contextual changes, the reactivation index in CA1 did not change accordingly, though the mice were able to detect the alteration in environmental inputs as indicated by their behaviors. This observation is consistent with previous findings that small changes in the environmental inputs may not be sufficient to cause remapping in the CA1 (Leutgeb et al., 2004). Similarly, it is found that lesions in CA1 do not affect the spatial discrimination ability of animals (Gilbert et al., 2001). Finally, other than providing an indication for memory recall, the behavioral readout is the result of neurological processes that are carried out by the entire brain, including regions other than the hippocampus. For example, amygdala is also involved in contextual fear conditioning and contributes to the freezing behavior during the re-exposure. Hence, the activity in the hippocampus may not necessarily be linearly correlated with behavior under all circumstances.

*4. The main conclusions of the paper rely on comparisons between 2 populations of activated neurons, using measures that are very different from each other. lacZ expression is used to measure long-term activation of neurons (i.e., in response to the first experience, days ago) whereas c-fos or zif268 is used to measure recent activity (for the 2nd experience). In CA1 lacZ is expressed in ∼6% of neurons whereas c-fos is expressed in ∼50% of neurons, nearly a 10x difference. In the DG zif268 is expressed at about twice the lacZ levels (12% vs 6%). The dramatic differences in activity-dependent expression of these two markers make it difficult to understand how they can be compared. In other words, it seems that they don't represent activity equally at all. Granted, this problem is somewhat tempered by the fact that activation and reactivation measurements are both calculated as % of cells expressing IEGs zif or fos (% of the total population vs % of the lacZ population). Clearly these two populations of cells are differentially activated in some cases and maybe it's the case that lacZ expression does not identify a qualitatively different type of activity, but just identifies fewer activated cells (e.g., because it has a higher threshold?). It would be more convincing if this was demonstrated. Right now there are factors that make one wonder if they might just be detecting different types of activity*.

As discussed earlier, the induced expression efficiency of the LacZ tag is significantly lower than that the endogenous IEG expression. We believe this is a general problem associated with the transgenic mice, since it is observed in different brain regions, including basolateral amygdala in the original study by Reijimer et al (2007), and CA1 and DG described here. Our interpretations are based on the assumption that LacZ tagged neurons are representative of the activated neuron population. As discussed above, supporting this assumption are the new data resulting from measuring the intensity of FOS staining in FOS+LacZ+ and FOS+LacZ- neurons in mice whose LacZ tagged and FOS labeled neurons were both responding to the same EE exposure (Figure 1–figure supplement 3). Realizing that the tagging efficiency is particularly low in CA1 (thus would most likely cause biases in this region), we quantified the activities by calculating the proportion of IEG+ cells in either the LacZ+ population or the total population instead of comparing the absolute numbers of IEG+ and IEG+LacZ+ cells to overcome the problem of the low tagging efficiency. Our results suggest that the CA1 neurons involved in memory encoding are preferentially activated by recall of the same memory trace, which is consistent with previous reports suggesting the role of CA1 in both encoding and retrieval (Riedel et al., 1999; Goshen et al., 2011). Similar to LacZ tagging, IEG labeling efficiency can vary depending on the marker used for examination. In the fear conditioning experiment 1, EGR1 and FOS labels about 14% and 7% activated DGCs in response to re-exposure, respectively. In the DG, the LacZ labeling efficiency is about two fold lower than that of the EGR1 labeling and is almost as high as that of the FOS labeling. Nevertheless, the reactivation indexes resulting from the analyses by these two different IEGs are similar (Figure 4), suggesting that using the percentage measures can help to normalize the difference in expression efficiencies of activity markers. However, it is realized that the possibility of LacZ tagging a specialized group of activated neurons (for example, the most active neurons) cannot be formally ruled out and we discuss this caveat in the revised manuscript.

*5. Due to the slow onset of the tTA-tetO system, the authors remove Dox 3d before subjecting mice to experiences for activating neuronal ensembles. Thus, there is a significant amount of time (and therefore experience) during which DG neurons could be activated (and express lacZ) based on home cage experiences. It is shown that home cage lacZ expression is lower than context-induced activation but how do we know that, in the dual context experiments, lacZ expression is due to the brief context pre-exposure rather than home cage activity? This could be addressed by showing that there is very little IEG expression in lacZ cells when the mice are taken off Dox but not exposed to the first context. A related issue lies with presentation of the experimental design. In the figures (e.g., Figure 1b, Figure 3a, and Figure 5a) the “on Dox” arrows extend significantly into the portion of the experimental timeline where the animals are not on Doxycicline. This gives the impression that Dox is removed at the time of the 2nd experience when in fact it is removed much earlier, in the home cage situation*.

Neither LacZ nor IEG expression is induced only by the pre-exposed context or the re-exposed context because home cage activity is an unavoidable part of whole experiences at both pre-exposure and re-exposure. We performed a new experiment to address this concern by taking mice off dox followed by exposing one group to context A (ctxA group) and keeping the other group in their home cage (HC group) before the re-exposure to context A. As predicted, we found that significantly fewer LacZ+EGR1 double positive neurons in LacZ-positive population (i.e., reactivation rate) in HC mice compared to ctxA mice. The new data are reported in Figure 5. In addition, the concern could also be addressed by a comparison between preA and preC mice. Both groups not only have the same home cage experience but also have equivalent numbers of LacZ positive neurons induced in CA1; however, only preA but not preC mice can recall the experience of being exposed to context A. If home cage activity had played a key role, the effect of differential context pre-exposure would have been masked. Given that our results showed a clear difference between preA and preC mice, a significant number of the LacZ and IEG positive neurons should respond to context pre-exposure and re-exposure, respectively. We have also revised figures (Figures 1, 3, and 6) according to the reviewers' suggestion to make the illustration clearer and we point out in the figure legends that the dox treatment is illustrated by the blue shading.

*6. It is not clear why the immediate shock contextual fear paradigm is used. Most previous studies that have examined the role of the DG in pattern separation have looked at activity in response to exploration of environments that are not associated with reward or shock. This doesn't necessarily have to be the case but for the current study the additional experiences associated with the learning process seem to confound interpretations. For example, to what extent does the 2nd experience activate neurons due to emotional content (recalling a memory of being shocked) vs. due to the fact that it's simply a different spatial environment? Also, why was the shock floor covered during pre-exposure and test but not during the shock? Why cover it at all? Thus, even animals that were exposed to the same context for both experiences (in order to see if the same population of DG neurons encodes re-experience of the same context) were also exposed to an additional similar context (one without the covered floor). So, what memory are the animals really recalling? Everything would be much cleaner if the mice were just exposed to one context and then a second context, with no intervening shocks or altered environments*.

The sequential exposure of animals to two contexts (same or different) is a commonly used design for this type of experiment; however, whether retrieval of the memories of the first exposure occurs during the second exposure cannot be determined using such a protocol. We therefore chose a contextual fear-conditioning paradigm to obtain an indication for recall of the pre-exposure experience during the second exposure. The immediate shock contextual fear paradigm is used because there were no extra stimuli (i.e., those elicited shock) in the pre-exposure compared to the re-exposure. The floor is covered during the pre-exposure because the wired floor is a very strong cue and will induce high freezing behavior in a similar context (i.e., context B). This rationale is now included in the Material and methods. Given that it is difficult for animals to form the context-shock association if shock is delivered soon after animals are introduced to the context (a.k.a. immediate shock deficit (Fanselow, 1990)), freezing behavior during the re-exposure is mainly a reflection of the memory of the pre-exposed context, though the recall of shock context may also occur. It is shown by both previous studies (Rudy and O'Reilly, 2001) and our data (Figure 3b) that animals without pre-exposure or pre-exposed to an irrelevant context do not display freezing behaviors to the conditioned context. However, we agree with the reviewers that the emotional value of the context altered after the immediate-shock; therefore, we performed a new experiment to address this issue by exposing mice sequentially to two contexts (either same or different) without the immediate-shock step (ctxA/A vs ctxA/C mice in Figure 5). The data are presented in Figure 5. Results of this experiment are similar to that of the contextual fear conditioning experiment (Figure 3), suggesting that changes in the emotional value of the context do not significantly influence the reactivation pattern of neurons in CA1 and the DG.

*7. The cell quantification methods are poorly described. First, what is meant by “total number of DGCs (dentate granule cells)” and why was it, along with density, calculated? Some mention of calculations is made but not described so it may become clear if described better. Perhaps, since the proportion of DG neurons that are activated is quite low, the authors manually counted all IEG and lacZ expressing cells but did not want to manually count all the unactivated cells (NeuN and DAPI cells). So there was some calculation of local DG neuron density, which was then used to extrapolate to estimate the actual number of DG neurons that was sampled. Is this correct? In any case, there is some concern about these methods since this density-to-total-number-of-DG-neurons conversion can introduce biases. This is significant because the activation rate is based on this number (IEG/total number of DG neurons) whereas the reactivation rate is not (both numerator and denominator are based on manual cell counts with no density conversions) and many of the key comparisons are between activation and reactivation rates. CA1 analyses are also based on manual cell counts so comparisons there are also problematic*.

Estimation was used in the quantification of the total number of DGCs in the images, which could affect the interpretation of the results. Prompted by the reviewers' suggestion, we developed another quantification method without estimation and re-analyzed the DG data in the contextual fear conditioning experiment 1 by quantifying the four types of cells – DAPI+ cells, EGR1+ cells, LacZ+ cells, and EGR1+LacZ+ cells – in a single plane in the z-stack of the DG images. The results yielded from this new quantification approach, presented in Figure 3d, f, and g, are similar to data resulting from the original method involving estimation (which are presented in Figure 3–figure supplement 2 in the revised manuscript). In addition, we also revised the description of the estimation methods to make it clearer and discussed our rationale, in response to the detailed comments from the reviewers.

Our original concern was that the LacZ+IEG+ double positive neurons were scarce in the DG and we thus quantified the entire z-stack of confocal images to increase our sample size. Due to the high density of neurons in the DG, the counting of total numbers of DGCs in the z-stack became very difficult (see DAPI image in Figure 3–figure supplement 1) and we thus used a compromise method to estimate DGC numbers by measuring the volumes of the DG region in the images. This method allows us to make legitimate comparison of activity rates between experimental groups in the DG, but, as pointed out by the reviewers, the comparison of the reactivation rates to the corresponding activation rates may be affected. We recognized that an over-estimation of the number of DGCs would not affect the data interpretation; however, under-estimation might result in overvalued activation rates. The data showed that the reactivation rate was almost half of the corresponding activation rate in preC or testB mice (Figure 3–figure supplement 2 and Figure 6), leading to the conclusion that distinctive population of DGCs is selected to represent a particular event. This interpretation would have been affected only if we had underestimated the DGC population by at least two fold, which is unlikely in our opinion. An underestimation of DGC population to a lesser extent would have resulted in the activation rate lower than the corresponding reactivation rate in the DG of preA or testA mice, leading to the interpretation that memory recall preferential reactivation of those DGCs involved in encoding the same memory (similar to CA1). Even in this case, the extent of the preferential reactivation in the DG would be still smaller compared to that of CA1. However, in light of the new analyses, these hypothetical situations are unlikely to happen because similar data were obtained using an alternative quantification method without estimation.

**References**

Alme, CB, Buzzetti, RA, Marrone, DF, Leutgeb, JK, Chawla, MK, Schaner, MJ, Bohanick, JD, et al. 2010. Hippocampal granule cells opt for early retirement. *Hippocampus*
**20**, 1109–23. doi: 10.1002/hipo.20810.

Fanselow, MS 1990. Factors governing one trial contextual conditioning. *Animal Learning and Behavior*
**18**, 264–70.

Gilbert, PE, Kesner, RP & Lee, I 2001. Dissociating hippocampal subregions: double dissociation between dentate gyrus and CA1. *Hippocampus*
**11**, 626–36. doi: 10.1002/hipo.1077.

Goshen, I, Brodsky, M, Prakash, R, Wallace, J, Gradinaru, V, Ramakrishnan, C & Deisseroth, K 2011. Dynamics of retrieval strategies for remote memories. *Cell*
**147**, 678–89. doi: 10.1016/j.cell.2011.09.033.

Leutgeb, JK, Leutgeb, S, Moser, MB & Moser, EI 2007. Pattern separation in the dentate gyrus and CA3 of the hippocampus. *Science*
**315**, 961–6. doi: 10.1126/science.1135801.

Leutgeb, S, Leutgeb, JK, Treves, A, Moser, MB & Moser, EI 2004. Distinct ensemble codes in hippocampal areas CA3 and CA1. *Science*
**305**, 1295–8. doi: 10.1126/science.1100265.

Mathews, EA, Morgenstern, NA, Piatti, VC, Zhao, C, Jessberger, S, Schinder, AF & Gage, FH 2010. A distinctive layering pattern of mouse dentate granule cells is generated by developmental and adult neurogenesis. *J Comp Neurol*
**518**, 4479–90. doi: 10.1002/cne.22489.

Neunuebel, JP & Knierim, JJ 2012. Spatial firing correlates of physiologically distinct cell types of the rat dentate gyrus. *J Neurosci*
**32**, 3848–58. doi: 10.1523/JNEUROSCI.6038-11.2012.

Reijmers, LG, Perkins, BL, Matsuo, N & Mayford, M 2007. Localization of a stable neural correlate of associative memory. *Science*
**317**, 1230–3. doi: 10.1126/science.1143839.

Riedel, G, Micheau, J, Lam, AG, Roloff, EL, Martin, SJ, Bridge, H, De Hoz, L, et al. 1999. Reversible neural inactivation reveals hippocampal participation in several memory processes. *Nat Neurosci*
**2**, 898–905. doi: 10.1038/13202.

Rudy, JW & O'reilly, RC 2001. Conjunctive representations, the hippocampus, and contextual fear conditioning. *Cogn Affect Behav Neurosci*
**1**, 66–82.